# Do better ImageNet classifiers assess perceptual similarity better?

**Manoj Kumar**  *mechcoder@google.com*

**Neil Houlsby**  *neilhoulsby@google.com*

**Nal Kalchbrenner**  *nalk@google.com*

**Ekin D. Cubuk**  *cubuk@google.com*

*Google Research, Brain Team*

**Reviewed on OpenReview:** *https://openreview.net/forum?id=qrGKGZZvHO*

## Abstract

Perceptual distances between images, as measured in the space of pre-trained deep features, have outperformed prior low-level, pixel-based metrics on assessing perceptual similarity. While the capabilities of older and less accurate models such as AlexNet and VGG to capture perceptual similarity are well known, modern and more accurate models are less studied. In this paper, we present a large-scale empirical study to assess how well ImageNet classifiers perform on perceptual similarity. First, we observe a inverse correlation between ImageNet accuracy and Perceptual Scores of modern networks such as ResNets, EfficientNets, and Vision Transformers: that is better classifiers achieve worse Perceptual Scores. Then, we examine the ImageNet accuracy/Perceptual Score relationship on varying the depth, width, number of training steps, weight decay, label smoothing, and dropout. Higher accuracy improves Perceptual Score up to a certain point, but we uncover a Pareto frontier between accuracies and Perceptual Score in the mid-to-high accuracy regime. We explore this relationship further using a number of plausible hypotheses such as distortion invariance, spatial frequency sensitivity, and alternative perceptual functions. Interestingly we discover shallow ResNets and ResNets trained for less than 5 epochs only on ImageNet, whose emergent Perceptual Score matches the prior best networks trained directly on supervised human perceptual judgements.

## 1 Introduction

ImageNet (Russakovsky et al., 2015) is the cornerstone of modern supervised learning and has enabled significant progress in computer vision. Features learnt via training on ImageNet transfer well to a number of downstream tasks (Carreira & Zisserman, 2017; Zhai et al., 2019; Huang et al., 2017), making ImageNet pretraining a standard recipe. Further, better accuracy on ImageNet usually implies better performance on a diverse set of downstream tasks such as robustness to common corruptions (Orhan, 2019; Xie et al., 2020b; Taori et al., 2020; Radford et al., 2021), adversarial robustness (Cubuk et al., 2017; Xie et al., 2020a), out-of-distribution generalization (Recht et al., 2019; Andreassen et al., 2021; Miller et al., 2021), transfer learning on smaller classification datasets (Kornblith et al., 2019), pose estimation (Mathis et al., 2021), domain adaptation (Zhang & Davison, 2020), object detection and segmentation (Zoph et al., 2020), and for predicting neural recordings and behaviors of primates on object recognition tasks (Schrimpf et al., 2020).

As a remarkable side effect, ImageNet models can also capture a notion of similarity identical to humans, known as perceptual similarity. Designing distance metrics that correspond to human judgements is a well

established problem in computer vision, and a number of low-level metrics (Zhang et al., 2011; Wang et al., 2004; Mantiuk et al., 2011) have been introduced for this purpose. The first generation of ImageNet classifiers: AlexNet (Krizhevsky et al., 2012), VGG (Simonyan & Zisserman, 2014), and SqueezeNet (Iandola et al., 2016) can all measure perceptual similarity termed as Perceptual Scores (PS), as an emergent property, in a way that outperforms all prior pixel-level metrics and correlates better with human judgement (Zhang et al., 2018).

In this paper, we are motivated by the following questions: Considering ImageNet classification has progressed significantly since then, can we obtain a better perceptual similarity metric by using a better classifier directly? Since modern neural network training involves a large number of hyperparameters, are there design choices that can improve a classifier's perceptual similarity? Are there latent factors that govern the relationship between ImageNet accuracy and perceptual similarity?

We perform a suite of experiments on BAPPS, a large dataset of human-evaluated perceptual judgements (Zhang et al., 2018). To the best of our knowledge, our work is the first empirical study to present a systematic investigation and rigorous deep dive into the relationship between ImageNet accuracy and perceptual similarity. We study the ImageNet accuracy/PS interplay of a wide variety of networks across many combinations of architectures and hyperparameters. Given the increasing interest in analyzing how representations of ImageNet classifiers transfer to other domains, our work adds another direction to this literature.

Fig. 1 displays the ImageNet accuracy and PS of every ImageNet classifier in our study. While Zhang et al. (2018), show a positive correlation between PS and ImageNet accuracy, we observe that this holds only in the low-accuracy regime. In the mid-to-high accuracy regime, we uncover a counter-intuitive Pareto frontier between accuracy and PS. Modern networks, EfficientNets, Vision Transformers and ResNets, lie to the right, obtaining high accuracies and low PS. Contrary to prevailing evidence that suggests models with high validation accuracies on ImageNet are likely to transfer better to other domains, we find that representations from underfit ImageNet models with modest validation accuracies achieve the best PS. Our experiments further suggest that attaining the best PS is somewhat architecture agnostic. For example, ResNets trained with large amounts of weight decay or early stopped within a few epochs of training can match or outperform the PS of AlexNet and VGG.

Finally, we investigate if there are latent factors that govern the relationship between ImageNet accuracy and PS using the following hypotheses. Does the inverse-U persist with global perceptual functions? Are low PS models less sensitive to distortions? What are the contributions of skip connections in modern architectures to decreased PS, if any? Do lower layers of better classifiers have a higher PS than higher layers? What is the impact of ImageNet class granularity on PS? While the causatory latent factor that governs the relationship between accuracy and perceptual similarity remains unclear, our paper opens the door to further understanding of this phenomenon.

A summary of our experiments are:

- We systematically evaluate the PS of modern "out-of-the-box" networks, ResNets, EfficientNets and Vision Transformers.

- We study the variation of PS (and accuracy) as a function of width, depth, number of training steps, weight decay, label smoothing and dropout. Our large-scale study consists of 722 different ImageNet networks across the cross product of these 7 different hyperparameters and 5 architectures.

- We explore the relationship between ImageNet accuracy and PS further using spatial frequency sensitivity, invariance to distortions, class granularity, and improved global perceptual functions.

Our empirical study leads to the following surprising results:

- While modern classifiers outperform prior pixel-based metrics in PS, they under-perform moderate classifiers like AlexNet.

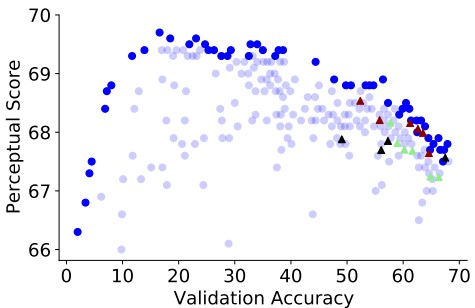

Figure 1: We discover a Pareto frontier (marked in dark blue) between Perceptual Scores (Zhang et al., 2018) on the 64 × 64 BAPPS Dataset (y-axis) and ImageNet 64 × 64 validation accuracies (x-axis). Each blue dot represents an ImageNet classifier. Better ImageNet classifiers achieve better Perceptual Scores up to a certain point. Beyond this point, improving on accuracy hurts Perceptual Score. Modern networks, EfficientNets (▲), Vision Transformers (▲) and ResNets (▲) lie far to the right of the point of optimal Perceptual Score, achieving high accuracy and lower Perceptual Scores. The best Perceptual Scores are attained by classifiers with moderate accuracy (20.0-40.0). The lowest and highest Perceptual Score observed in prior networks are 64.3 (using random weights, i.e. an untrained network) and 68.9 (AlexNet) respectively.

- Modern ImageNet models that are much shallower, narrower, early-stopped within a few epochs of training and trained with larger values of weight decay attain significantly higher PS than their out-of-the-box counterparts.

- In all of our hyperparameter sweeps with the exception of label smoothing and dropout, we discover an unexpected and previously unobserved tradeoff between accuracy and PS. In each hyperparameter sweep, there exists an optimal accuracy up to which improving accuracy improves PS. This optimum is fairly low and is attained quite early in the hyperparameter sweep. Beyond this point, better classifiers achieve worse PS.

- Perceptual functions that rely on global image representations such as style achieve better PS than per-pixel perceptual functions.

- We do not find a correlation between PS of models and their sensitivity to distortions. Further, low Perceptual Score models are not necessarily more reliant on high-frequency spatial information for classification as compared to high PS models.

- High-level features found in the latter layers of residual networks, have better PS than low-level features found in the earlier layers.

- Finally, we find particularly shallow, early-stopped ResNets trained only on ImageNet that attain an emergent Perceptual Score of 70.2. This matches the best Perceptual Scores across prior networks which were trained directly on BAPPS to match human judgements.

## 2 Related Work

There has been a rich body of work dedicated to analyzing the transfer of pretrained ImageNet networks to various downstream tasks. Kornblith et al. (2019) show that transfer learning performance is highly correlated with ImageNet top-1 accuracy. Taori et al. (2020); Miller et al. (2021) show that improvements in accuracy on ImageNet consistently results in improvements on test datasets with distribution shift. Djolonga et al. (2021) show a positive correlation between ImageNet transfer performance and out-of-distribution robustness. While Geirhos et al. (2018) demonstrate that ImageNet-trained models have a stronger bias towards using texture cues compared to humans, Hermann et al. (2019) show that among high-performing models, shape-bias is correlated with ImageNet accuracy. Schrimpf et al. (2020) found that CNNs that achieve a higher accuracy

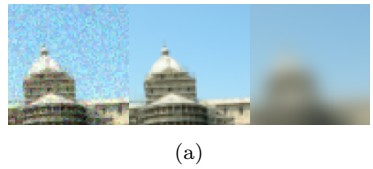 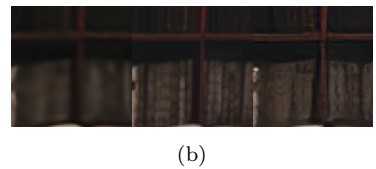 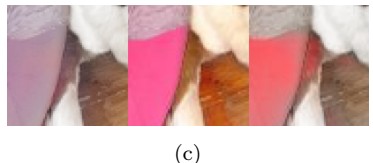

(a)                    (b)                    (c)

Figure 2: Three sample triplets from the BAPPS Dataset corresponding to the traditional (left), super-resolution (center) and color (right) distortion families. Each human rater provides a binary label, that indicates which of the two patches are closer to the center patch, that is label 0 if the left patch is closer. The mean rating for these three triplets across 5 raters are 0.0, 1.0 and 0.8, respectively.

on ImageNet are better at predicting neural recordings of primates when executing object recognition tasks, but the positive correlation was weaker for high-accuracy models. While better ImageNet classifiers transfer better on the above tasks, we are the first to observe a negative correlation between ImageNet classifiers and their inherent ability to capture image similarity. While we do expect that larger models with more capacity can attain better PS when trained directly on BAPPS, we use BAPPS solely to evaluate the emergent perceptual properties of ImageNet-trained classifiers.

An orthogonal line of work, involves incorporating "perceptual losses" that minimize the distance between intermediate features of a ground-truth and generated image to drive photorealistic synthesis in style transfer (Gatys et al., 2015), (Li et al., 2017), super-resolution (Johnson et al., 2016), and conditional image synthesis (Dosovitskiy & Brox, 2016), (Chen & Koltun, 2017). Following the investigation done in (Zhang et al., 2018), state-of-the-art models in image-synthesis domains such as super-resolution (Lugmayr et al., 2020; 2021), (Bhat et al., 2021), image inpainting (Nazeri et al., 2019), (Zhao et al., 2021), high-resolution image synthesis (Esser et al., 2021), (Karras et al., 2019), (Karras et al., 2020), deblurring (Kupyn et al., 2019), (Zhang et al., 2020), image-to-image translation (Richardson et al., 2021), (Lee et al., 2018b), video generation (Babaeizadeh et al., 2021), (Franceschi et al., 2020), (Wu et al., 2021), (Lee et al., 2018a), neural radiance fields (Martin-Brualla et al., 2021), (Mildenhall et al., 2020), view synthesis (Wang et al., 2021b), (Riegler & Koltun, 2020), (Tucker & Snavely, 2020), (Wiles et al., 2020), and others (Lai et al., 2018), (Liu et al., 2021) use perceptual distance as either an auxiliary training loss to improve image synthesis or as an evaluation metric to assess synthesized image quality. While there has been some anecdotal evidence regarding the greater efficacy of VGG features over ResNet-50 for style losses (Nakano, 2019), there has been no systematic investigation assessing the relationship between ImageNet accuracy and perceptual similarity. In this paper, we exclusively focus on uncovering the relationship between the accuracy of ImageNet models and their inherent ability to capture perceptual similarity and not on applications of perceptual losses to downstream tasks.

## 3 Background

### 3.1 The BAPPS Dataset

The BAPPS Dataset (Zhang et al., 2018) is a dataset of 161k patches derived by applying exclusively low-level distortions to the MIT-Adobe 5k dataset (Bychkovsky et al., 2011) for training and the RAISE1k dataset (Dang-Nguyen et al., 2015) for validation. (Zhang et al., 2018) consider 6 distortion families namely:

- Traditional Distortions: Random noise, blurring, spatial shifts, corruptions and compression artifacts. (1 family.)

- CNN-based Distortions: Distortions created by CNN-based autoencoders trained on autoencoding, denoising, colorization and superresolution. (1 family.)

- Outputs of Real algorithms: Outputs from state-of-the-art frame interpolation, video deblurring, colorization and superresolution models. The distortions created by each class of models is treated as a separate family. (4 families.)

Table 2 in Zhang et al. (2018) contains a comprehensive list of distortions. The train set consists of the traditional and CNN-based distortions and the validation set contains all 6 families. Given a family of distortions and a set of reference images, Zhang et al. (2018) generate the BAPPS dataset as follows. They select a reference patch $x$ and then apply two distortions at random to generate the target patches $x_0$ and $x_1$. They record the binary response of a human, indicating which of the target patches is closer to the reference patch. For a given image triplet, $(x_0, x_1, x)$, $p$ is the average of 2 and 5 human responses on the train and validation set respectively. Fig. 2 displays 3 sample image triplets from the BAPPS Dataset.

### 3.2 Perceptual Score

We first define the PS of a network, for which we adopt the "2AFC" scoring protocol (Zhang et al., 2018). First, let $x_0$ and $x_1$ denote two images. Let $\tilde{y_0}^l$ and $\tilde{y_1}^l$ be the feature maps for $x_0$ and $x_1$ at the $l$th layer of a network, normalized across the channel dimension. The perceptual similarity function $d(x_0, x_1)$ is defined as:

$$d(x_0, x_1) = \sum_{l \in \mathcal{L}} \frac{1}{H_l W_l} \sum_{h,w} ||\tilde{y_0}^l_{h,w} - \tilde{y_1}^l_{h,w}||^2 \tag{1}$$

where $H_l$ and $W_l$ denote the height and width of the feature maps at layer $l$, respectively. $\mathcal{L}$ denotes the subset of layers which are used in the perceptual similarity; this subset is architecture specific. Given a reference image $x$ and two target images $x_0$ and $x_1$, BAPPS (Zhang et al., 2018) provides a ground truth soft-label $p$. $p$ can be interpreted as the probability a human rater would rate $x_1$ as more similar to $x$ than $x_0$.

For a neural network with distances $d_0 = d(x, x_0)$ and $d_1 = d(x, x_1)$, from (Zhang et al., 2018) we define the PS $s(d_0, d_1)$ to be the following value times 100:

$$p \mathbb{1}[d_0 > d_1] + (1 - p) \mathbb{1}[d_1 > d_0] + 0.5 \mathbb{1}[d_0 = d_1] \tag{2}$$

**Dynamic Range of PS.** Unlike accuracy that can vary from 0 to 100, PS has a narrow dynamic range. To provide intuition on this dynamic range, we plot the PS for ground truth ratings $p$ from BAPPS and simulated combinations of $(d_0, d_1)$ dependent on $p$. Let $\tilde{p} = \mathbb{1}[p > 0.5]$, i.e we convert the human ratings into binary labels. If $\tilde{p} = 1$, we sample $d_0$ and $d_1$ from truncated normal distributions, i.e $d_0 \sim \phi(1, \sigma, 0, 1), d_1 \sim \phi(0, \sigma, 0, 1)$ where 0 and 1 are the lower and upper bounds. Conversely if $\tilde{p} = 0$, we sample $d_0 \sim \phi(0, \sigma, 0, 1), d_1 \sim \phi(1, \sigma, 0, 1)$. In Fig. 3, $\sigma$ models the noise in predicting $d$, as $\sigma$ is increased, the distances $(d_0, d_1)$ have a higher chance of being misaligned with $p$, and as expected PS smoothly decreases from the upper bound ($\sim 0.8$) to random choice (0.5).

On the actual BAPPS dataset, the previous best PS for low-level metrics was attained by FSIMc (Zhang et al., 2011), a low-level hand crafted matching function with a value of 63.8. The previously reported lower and upper PS bounds for ImageNet networks (Zhang et al., 2018) were values of 64.3 and 68.9.

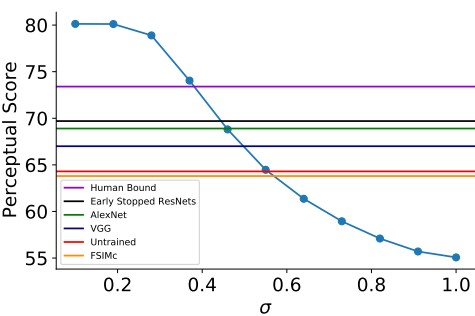

Figure 3: The blue line depicts PS on simulated distances and real labels from BAPPS. $\sigma$ models the noise in predicting the simulated distances from the real labels. The green and red horizontal lines denote the best (AlexNet) and worst PS (untrained) obtained with ImageNet networks. The orange line denotes the best PS obtained with low-level metrics (FSIMc). Early-stopped ResNets obtain the best PS score in this study.

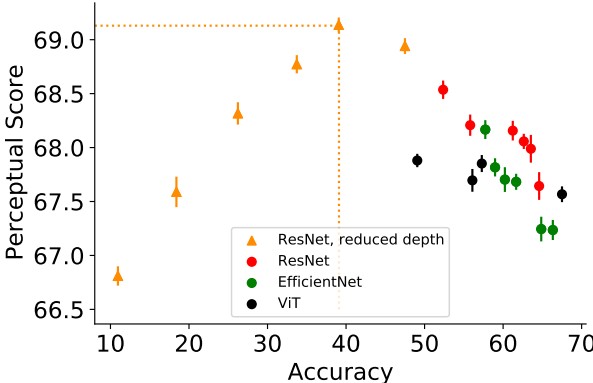

Figure 4: Perceptual Scores (Zhang et al., 2018) on the $64 \times 64$ BAPPS dataset as a function of ImageNet $64 \times 64$ validation accuracies. Each point is the average across 5 runs. Error bars on both the axes are the corresponding standard errors (variance in accuracy is sufficiently small that error bars in the x-direction are barely visible). Across the out-of-the-box ResNets, EfficientNets and ViTs (circles), there exists a negative correlation between Perceptual Score and accuracy. On reducing the depth even further, (ResNet-reduced depth, triangles) we uncover an "optimal accuracy threshold". Below this threshold, Perceptual Scores correlate with accuracy and above it Perceptual Scores decrease. Among the architectures in this plot, ResNet-6 attains this optimal accuracy threshold and has the highest Perceptual Score.

These were achieved by a randomly initialized network and AlexNet (Krizhevsky et al., 2012) respectively. VGG obtains a PS of 67.0. The human upper bound is 73.9, reflective of the entropy across human raters. Across all our experiments, we cover almost the entire prior dynamic PS range of trained networks, with the PS of our networks ranging from 65.0 to 69.7.

## 4 Experimental Setup

**Architectures.** We study two of the most popular classes of vision architectures: Convolutional Neural Networks (CNNs) and Transformers. As representative CNNs, we train ResNets (He et al., 2016) and EfficientNets (Tan & Le, 2019). For EfficientNets, we use model variants B0 to B5 (Tan & Le, 2019). For ResNets, we train networks with depths ranging from 6 to 200 layers. For Vision Transformers (ViT) (Dosovitskiy et al., 2020), we use the Base and Large variants, with patch sizes of 8 and 4, leading to four models (ViT-B/8, ViT-B/4, ViT-L/8 and ViT-L/4). We use smaller patch sizes than is common because we use lower resolution images (see the following Training section). However, any patch size smaller than 4 leads to poor generalization.

**Training.** We train our networks on ImageNet at a resolution of $64 \times 64$ and report their accuracies on the ImageNet validation set and PS on the BAPPS validation set. ImageNet $64 \times 64$ provides a cleaner test bed instead of the standard high-resolution ($224 \times 224$ and above) for the following reasons.

BAPPS is constructed using $64 \times 64$ images so that human raters can focus on low-level local changes as opposed to high-level semantic differences. ImageNet models pre-trained on high-resolution images classify $64 \times 64$ images poorly. Such models are sub-optimal for our analysis, because we are interested in the perceptual properties of classifiers that generalize reasonably well. An alternative option would be to resize $64 \times 64$ images to high-resolution images using linear interpolations. However, this can have the adverse side effect of blurring out or removing distortions from BAPPS images. Nonetheless, in Appendix A, we observe similar phenomena on models trained with high resolution images, with significantly worse PS.

For all networks, we apply only random crops and flips and disable all other augmentations. We use the recommended hyperparameter settings given by the corresponding open-sourced training code. For the

models where validation accuracy decreases during training, EffNet-B4 and EffNet-B5, we stop training just before this happens.

**Representations.** In prior work, Zhang et al. (2018) compute the PS of VGG using the outputs of every 2x2 max pooling layer, leading to 5 features in total. For the shallower AlexNet architecture, they employed the output of each of the 5 convolutions. Here, we describe the representations we use to compute the PS of modern networks, in this work.

Similar to VGG, for ResNets and EfficientNets, we use the outputs of the 4 reduction stages, where the spatial dimensions are reduced 2x via strided convolutions. Specifically, we obtain four 2D representations of resolution $16 \times 16$, $8 \times 8$, $4 \times 4$, and $2 \times 2$.

For Vision Transformers, we use the global CLS representation of the image at the output of every encoder block. Our initial experiments on using features at the output of every layer instead of every block or reduction stage, as computed in AlexNet lead to worse PS across all architectures.

## 5 ImageNet accuracy versus Perceptual Score

We train networks with their default hyperparameters five times and report the mean accuracy and PS ($64 \times 64$) with error bars in Fig. 4. (Appendix M contains a list of default hyperparameters). All modern networks (red, green, and black) in Fig. 4 obtain a higher PS than FSIMc (63.8) and randomly initialized networks (64.3). Surprisingly though, representations of better ImageNet networks perform consistently worse. For example, ResNet-18 which achieves the best PS of 68.5 has a modest accuracy of 52.0% ($64 \times 64$). EfficientNet B5 which has the worst PS of 67.2 achieves a much higher accuracy of 66.3%. Additionally, all networks perform worse than AlexNet (68.9).

Next, we push this observation to the limits. Our next experiment hopes to identify a much smaller network that achieves improved PS at the expense of extremely low accuracy. In Fig. 4, we report the accuracies and PS obtained by shallow ResNets with depths from 2 to 10 (ResNet, reduced depth). ResNet-10 and ResNet-6 obtain improved PS with reduced accuracies. However, the ResNets with a depth smaller than 6 incur losses in both accuracy and PS. Our results indicate that better classifiers produce better perceptual representations up until a certain "optimal accuracy". Above this optimal accuracy, ImageNet classifiers trade off better accuracies with worse PS. This phenomenon leads to some interesting observations: 1) ResNet-6 happens to achieve this optimal accuracy with a PS of 69.1, outperforming AlexNet. 2) Beyond the optimal accuracy, there is a strong inverse correlation between accuracy and PS (coefficient = -0.84). 3) ResNet-200 and ResNet-3 achieve similar PS, while having a an accuracy difference of 45%.

## 6 How general is this relationship?

Section 5 indicates that an inverse-U relationship exists between accuracy and PS on varying the depth of residual networks. In this section, we attempt to generalize this relationship as a function of other implicit hyperparameters such as layer composition, width, depth and training hyperparameters of the considered architectures. In particular, we assess this relationship between accuracy and PS as a function of hyperparameters via 1D sweeps. We vary a single hyperparameter of a network (e.g. width) along a 1D grid which changes the network's accuracy and as an effect, the corresponding PS. In Appendix H, for all our sweeps, the validation accuracy does not decrease during training, indicating none of the networks overfit on the ImageNet train set. For a given sweep, we provide two sets of plots: 1) Scatter plots between PS/accuracy. 2) PS against the hyperparameter values. We choose 5 architectures: ResNets and ViTs with the best and worst PS, (ResNet-6, ResNet-200, ViT-B/8, ViT-L/4) plus a standard ResNet-50.

We sweep across the following hyperparamters: Number of training steps, network width, network depth, weight decay (Hanson & Pratt, 1988), dropout (Srivastava et al., 2014), and label smoothing (Szegedy et al., 2016b). We define $p_{max}$ to be the optimal PS and $a(p_{max})$ to be the accuracy at which $p_{max}$ is reached. We observe the inverse-U relationship across all these hyperparameters with the exception of label smoothing

and dropout. For these hyperparameter settings, we indicate $p_{max}$ and $a(p_{max})$ with dotted lines parallel to the x and y axes, respectively.

**Number of train epochs.** All networks exhibit the inverse-U shaped behaviour (Fig. 5a and 5c). $a(p_{max})$ is fairly consistent within a given family of architectures. ResNets have $a(p_{max})$ at low values between 15-25 % and ViTs have $a(p_{max})$ at moderate values between 40-50 %.

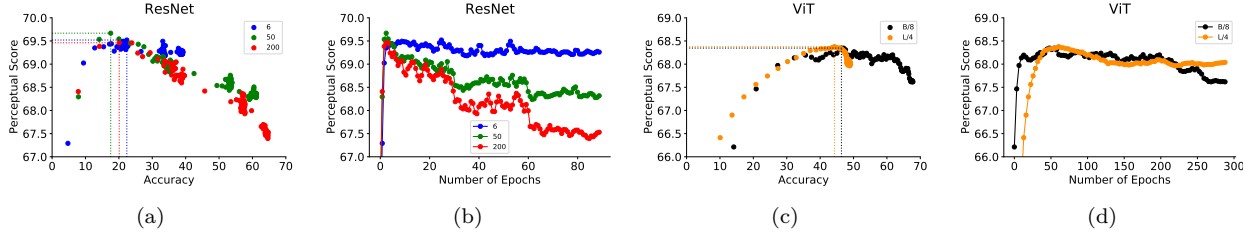

Figure 5: Figs. 5a and 5c show PS vs accuracy on varying train epochs in ResNets and ViTs. Figs. 5b and 5d depict PS as a function of train epochs.

PS peaks early during training across all networks (Fig. 5d and 5b). ResNet-50 and ResNet-200 peak at the first few epochs of training ($p_{max} = 69.7$) while the ViT models peak at lower values ($p_{max} = 68.4$) and later around 60 epochs. After the peak, PS of better classifiers decrease more drastically. ResNets are trained with a learning rate schedule that causes a step-wise increase in accuracy as a function of training steps. Interestingly, in Fig. 5b, they exhibit a step-wise decrease in PS that matches this step-wise accuracy increase.

**Width and Depth.** In Fig. 6a, we see that ResNet-6 achieves $p_{max} = 69.6$ at $a(p_{max}) = 20\%$ while ResNet-200 has $p_{max} = 67.6$ at $a(p_{max}) = 65\%$. As the model capacity is increased from ResNet-6 to ResNet-200, the peak shifts from the top left to the bottom right. Compound scaling is a model scaling technique (Tan & Le, 2019; Szegedy et al., 2016a) that improves model accuracy efficiently by scaling the width and depth of a network together. We observe a peculiar "inverse compound scaling" phenomena; depth and width of networks have to be scaled down simultaneously to improve on PS significantly.

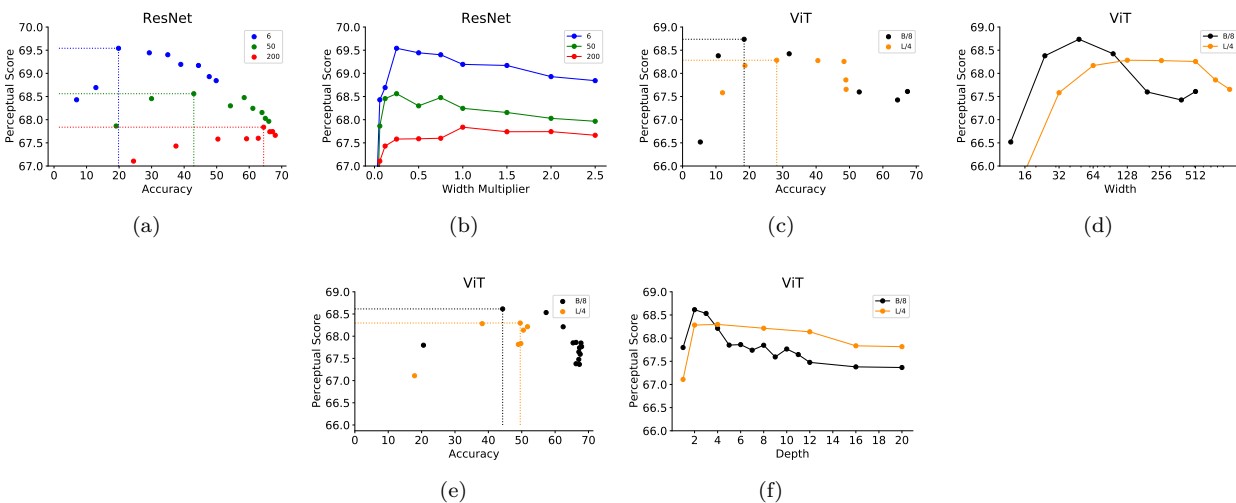

Figure 6: Figs. 6a, 6c and 6e show PS vs accuracy on varying width in ResNets, width in ViTs and depth in ViTs. Figs. 6b, 6d and 6f depict PS as a function of width in ResNets, width in ViTs and depth in ViTs.

$a(p_{max})$ on varying the width and depth of ViTs are at 20% to 30% and 45-50%, respectively. (Figs. 6e, 6c). A shallow ViT model of depth 2 gets close to 40% accuracy. Hence, there might just not be enough points between 20% to 40% in Fig. 6e. This could explain the shift of $a(p_{max})$ to the right in Fig. 6e when compared to Fig. 6c.

Shallower and narrower architectures perform better as shown in Figs. 6d, 6f, and 6b. The optimal width of ViT-B/8 and ViT-L/4 are 6 and 12% of their default widths while their optimal depths are just 2 transformer blocks. ResNet-6 and ResNet-50 exhibit similar properties with the optimal width being 25% of their original widths. ResNet-200 is the outlier with a small peak at its original width.

**Central Crop.** Modern networks employ random crops of high-resolution rectangular images during training. This artificially increases the effective quantity of training data available to the network. Replacing random crops with central crops has been shown to increase shape bias (Hermann et al., 2019) and reduce the discrepancy of object scales between training and testing (Touvron et al., 2019).

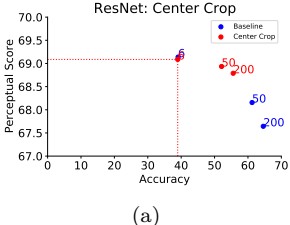 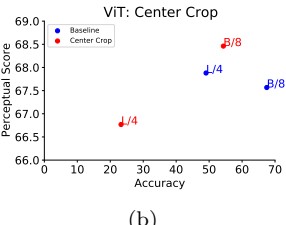

Figure 7: Figs. 7a and 7b show PS vs accuracy when random crop is replaced with central crop in ResNets and ViTs.

Fig. 6 shows the accuracies and PS of the 5 architectures trained with center crops. Each architecture moves towards the top-left with lower accuracies and higher PS. ViT-L/4 is the exception as it moves towards the bottom-right. It encounters a significant reduction in accuracy, lowering it below $a(p_{max})$, which could explain the decrease in its PS. All center-cropped architectures lie along an inverse-U, with ResNet-6 at the optimum.

**Weight Decay.** Figs. 8c and 8d display the impact of weight decay on PS. ViT-L/4 has minimal variation in accuracy as the weight decay factor is varied; so we omit ViT-L/4 from this study. ResNets and ViTs achieve their worst PS at the default weight decay around $10^{-4}$ and 0.3, respectively. The PS increases on either side of this optimum. This correlates with their changes in accuracy as a function of weight decay. ResNets and ViTs achieve their best accuracies at these weight decay values and decrease in either direction.

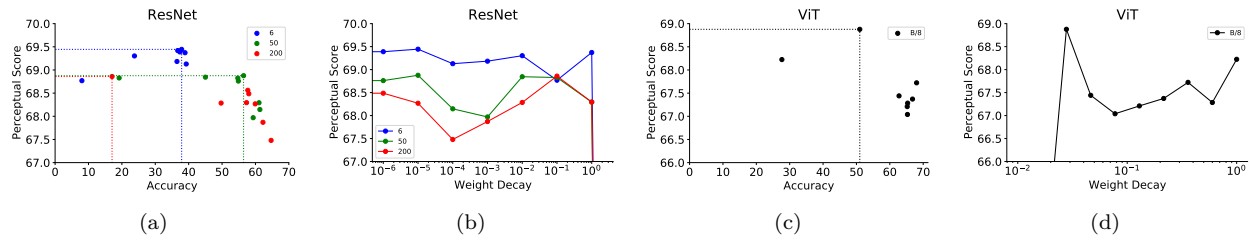

Figure 8: Figs 8a and 8b show PS vs accuracy on varying weight decay in ResNets and ViTs. Figs 8c and 8d depict PS as a function of weight decay.

**Label Smoothing and Dropout.** Across all our controlled settings, label smoothing and dropout are the only hyperparameters that decrease both PS and accuracy.

Varying label smoothing produces less drastic changes in accuracy as compared to other hyperparameters. In Fig. 9a, a very weak positive correlation exists between accuracy and PS for ResNet-50 and ResNet-200. In

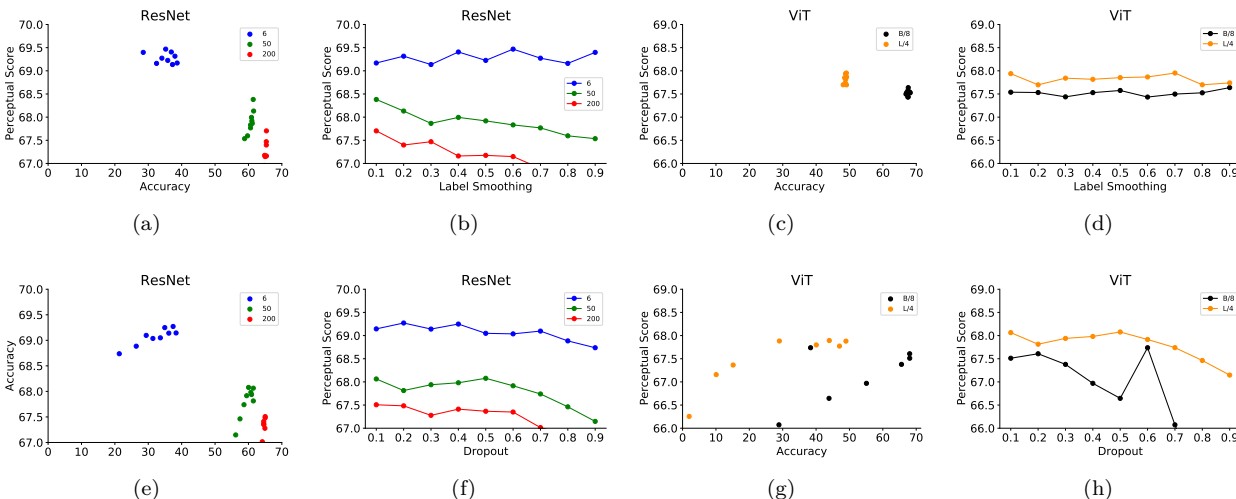

Figure 9: Figs 9a, 9c, 9e and 9g show PS vs accuracy on varying label smoothing in ResNets, label smoothing in ViTs, dropout in ResNets and dropout in ViTs. Figs 9b, 9d, 9f and 9h are the same graphs with PS as a function of the corresponding hyperparameters.

Fig. 9b, PS of ResNet-50 and ResNet 200 decrease with more label smoothing, while the Vision Transformers and ResNet-6 are almost invariant. Our results indicate that clean labels are necessary to obtain high PS. However, varying label smoothing does not change accuracy a great deal within each architecture class, so the dynamic range is insufficient to observe the inverse-U relationship.

In Figs. 9f and 9h the PS consistently decreases as a function of dropout. In Fig. 9e and Fig. 9g, it also correlates with accuracies. Curiously, dropout is the only factor to negatively influence both accuracy and PS simultaneously.

**Conclusion.** In Fig. 1, we plot the accuracy and PS from all our above experiments. While their exact relationship is architecture and hyperparameter dependent, we uncover a global Pareto frontier between PS and accuracies, see Fig. 1. Up to a certain peak, better classifiers achieve PS and beyond this peak, better accuracy hurts PS.

## 7   Scaling down improves Perceptual Scores

Table 1: Perceptual Score improves by scaling down ImageNet models. Each value denotes the improvement obtained by scaling down a model across a given hyperparameter over the model with default hyperparameters.

| Model | Default | Width | Depth | Weight Decay | Central Crop | Train Steps | Best |
|---|---|---|---|---|---|---|---|
| ResNet-6 | 69.1 | +0.4 | - | +0.3 | 0.0 | **+0.5** | 69.6 |
| ResNet-50 | 68.2 | +0.4 | - | +0.7 | +0.7 | **+1.5** | 69.7 |
| ResNet-200 | 67.6 | +0.2 | - | +1.3 | +1.2 | **+1.9** | 69.5 |
| ViT B/8 | 67.6 | +1.1 | +1.0 | **+1.3** | +0.9 | +1.1 | 68.9 |
| ViT L/4 | 67.9 | +0.4 | +0.4 | -0.1 | -1.1 | **+0.5** | 68.4 |

Our results in Section 6 prescribe a simple strategy to make an architecture's PS better: Scale down the model to reduce its accuracy till $a(p_{max})$.

Table 1 summarizes the improvements in PS obtained by scaling down each model across every hyperparameter. With the exception of ViT-L/4, across all architectures, early stopping yields the highest improvement

in PS. In addition, early stopping is the most efficient strategy as there is no need for an expensive grid search.

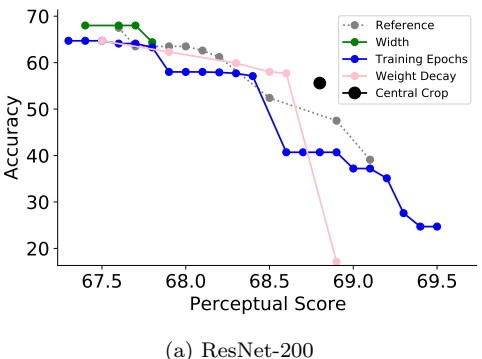

(a) ResNet-200

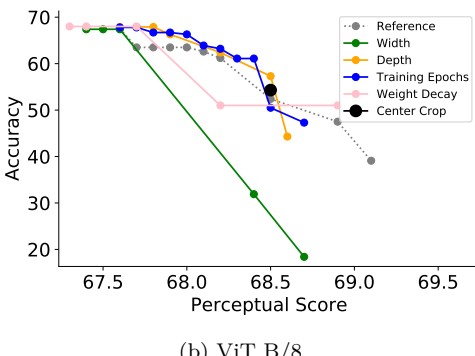

(b) ViT B/8

Figure 10: Each curve in a subplot showcases the validation accuracy vs Perceptual Score Pareto frontier for that architecture/hyperparameter combination. Each point on a line denotes the minimum accuracy loss achievable for a particular PS gain. The dashed gray line is the reference Pareto frontier obtained with out-of-the-box architectures.

**How much does PS improvement cost in terms of accuracy?** Fig. 10 shows the accuracy-PS Pareto frontier for ResNet-200 and ViT-B/8. Each point on the Pareto frontier denotes the maximum possible achievable accuracy for a given PS. The gray line is the reference Pareto-frontier obtained from training networks with their default settings. Except for Width + ViT-B/8 that lies below the reference Pareto frontier, all curves lie very close to the reference Pareto frontier. Early-stopping any of the architectures to improve their PS, as seen in Table 1, greatly reduces their accuracy.

# 8 Further Exploration: Reasons behind the inverse-U relationship

## 8.1 The inverse-U phenomenon persists with improved perceptual similarity functions

We first posit that the perceptual similarity function is suboptimal, and an alternative would not yield an inverse-U relationship.

The perceptual similarity function in (Zhang et al., 2018) averages per-pixel differences across the spatial dimensions of the image. This assumes a direct correspondence between pixels, which may not hold for warped, translated or rotated images. For a similarity function that compares global representations of images, the inverse-U relationship may no longer exist. We investigate two such functions in two different settings: 1) Out-of-the-box ResNets and EfficientNets. 2) ResNet-200 as a function of train steps.

**Style.** We adopt the style-loss function from the Neural Style Transfer work of (Gatys et al., 2015). Let $\tilde{y}_0^l$ and $\tilde{y}_1^l \in \mathbb{R}^{H_l \times W_l \times C_l}$ be the spatially normalized outputs of two images $x_0$ and $x_1$ at the $l$th layer of the network. We compute $G_0^l \in \mathbb{R}^{C_l \times C_l}$, the inter-channel cross-correlation matrix of $\tilde{y}_0^l$ and similarly $G_1^l$ for $\tilde{y}_1^l$. $G_0$ and $G_1$ also known as Gram matrices capture global information in $\tilde{y}_0^l$ and $\tilde{y}_1^l$ respectively. The style function is given by $\sum_l \frac{1}{C_l^2} ||G_0^l - G_1^l||_F^2$.

**Mean Pool.** Let $y_0^l$ and $y_1^l \in \mathbb{R}^{H_l \times W_l \times C_l}$ be the unnormalized outputs at the $l$th layer of the network. We spatially average them to global representations and then normalize across channels to obtain $\bar{y}_0^l$ and $\bar{y}_1^l \in \mathbb{R}^{C_l}$. The distance function is then given by $\sum_l \frac{1}{C_l} ||\bar{y}_0^l - \bar{y}_1^l||^2$.

In Figs. 11a and 11b, we present scatter plots between accuracy and PS with the style and mean pool similarity functions. Fig. 11a displays the accuracy and PS of ResNets and EfficientNets trained with their default hyperparameters. Each point in Fig. 11b represents a ResNet-200 model at a different epoch during the course of training.

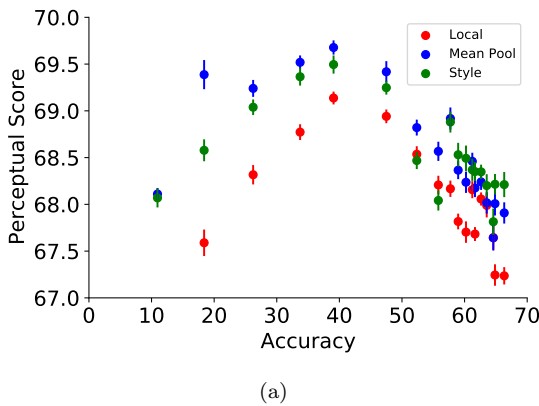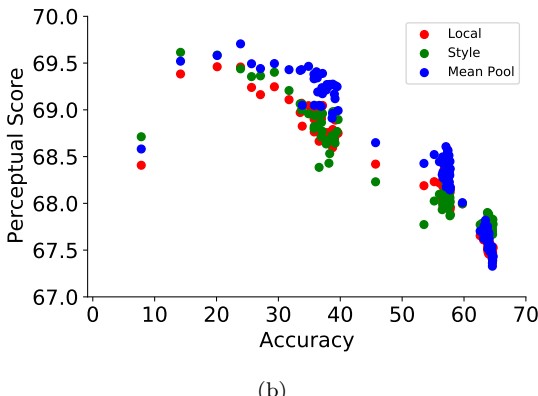

(a)                                                                                      (b)

Figure 11: Relationship between accuracy and PS via two alternate perceptual functions: **Mean Pool** and **Style**. Fig. 11a displays the accuracy and PS of ResNets and EfficientNets trained with their default hyperparameters. Each point in Fig. 11b represents a ResNet-200 model at a different epoch during the course of training.

Both functions yield better PS than the baseline ("Local") in (Zhang et al., 2018). In Fig. 11a ResNet-6 with its Mean Pool and Style variants outperform the baseline (local) 69.1 with scores of 69.7 and 69.5 respectively. In Fig. 11b for an early-stopped ResNet-200 model, the mean pool and style functions improve upon the baseline score of 69.5 with 69.8 and 69.7 respectively.

We additionally observe that the optimal early-stopped ResNet-6 from Table 1 further improves its performance with its mean pool variant achieving a PS of 70.2. This matches the best reported PS in (Zhang et al., 2018), where the AlexNet model is trained from scratch on the BAPPS train set. Note that none of our networks have seen the BAPPS train set during ImageNet training.

**However, although the improved perceptual functions attain better PS as compared to the baseline, the inverse-U correlation is still prominent**. Therefore, we can conclude that while the per-pixel comparison function is suboptimal, it is not the main cause of the inverse correlation between accuracy and PS.

**Learned linear layer on pretrained features.** Lastly we investigate what happens if the similarity function is learned on supervised data. Although the main goal of our paper is to assess the inherent perceptual properties of ImageNet models, we may also train a linear layer on top of pretrained ImageNet features to match supervised human judgements on BAPPS. The PS gap between ResNet-6 and ResNet-200 narrows down from 1.5 to 0.8, but even after training, the ResNet-6 still outperforms the ResNet-200. See Appendix C for more details.

## 8.2 Low PS models are not necessarily less sensitive to distortions

Here, we explore whether sensitivity to distortions is the common latent factor influencing both PS and accuracy. Intuitively, better networks will be less sensitive to the distortions in the BAPPS dataset, since the class will not change under these distortions. This intuition is supported by results that show that accuracy under distribution shifts (including artificial corruptions) correlates strongly with "clean" ImageNet accuracy (Taori et al., 2020). Therefore, if decreased sensitivity is related to poorer PS, due to inability to distinguish different class-preserving perturbations, then this could explain our observations.

From the BAPPS dataset, we retain only the examples, where the human raters unanimously agree that one of the target patches is closer to the reference patch than the other, i.e $p = 1.0$ or $p = 0.0$. For each such triplet $(x_0, x, x_1)$ where $p = 1.0$ or $p = 0.0$, we denote $x_f$ to be the farther patch and $x_n$ to be the nearer patch. Concretely, in Eq 2, when $p = 1.0, x_f = x_0, x_n = x_1$ or $p = 0.0, x_f = x_1, x_n = x_0$.

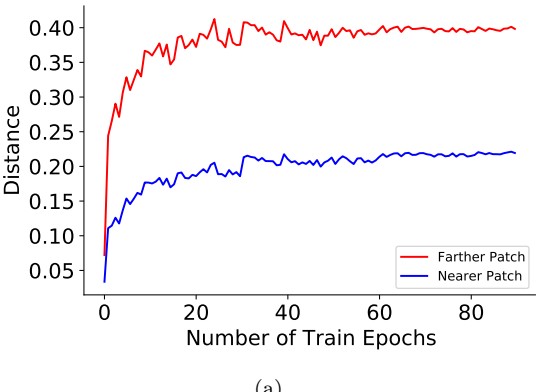 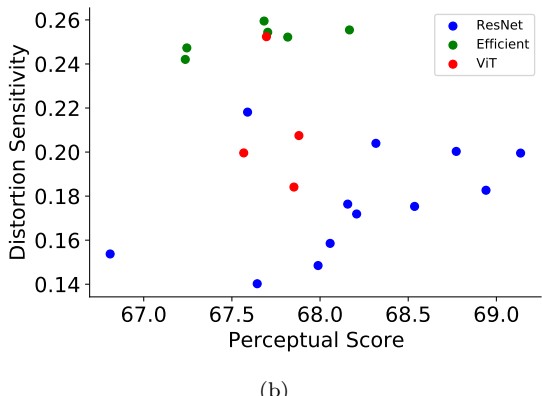

(a)  (b)

Figure 12: In Fig. 12a, distance assigned to the nearer and farther patch for a ResNet-200 model as a function of train steps. Fig. 12b displays a scatter plot between the distortion sensitivity measure (See: 8.2) and accuracy of networks trained with their default hyperparameters.

We measure distortion sensitivity using the distance margin $\mathbb{E}_{x,x_f} d(x, x_f) - \mathbb{E}_{x,x_n} d(x, x_n)$. We expect this margin to be larger for a distortion sensitive network. In Fig. 12a, among out-of-the-box classification networks, there exists no positive correlation between distortion sensitivity and PS. As another experiment, in Fig. 12b, we plot $\mathbb{E}_{x,x_f} d(x, x_f)$ (Farther Patch) and $\mathbb{E}_{x,x_n} d(x, x_n)$ (Nearer Patch) as a function of training epochs (ResNet-200). From Fig. 5b, we know that PS decreases as a function of epochs after it reaches a peak. However in Fig. 12b, the distance margin between the farther and nearer patch remains fairly constant as a function of epochs. **Hence, low PS models are not necessarily less sensitive to distortions.**

## 8.3 Sub-optimal features are not a cause of the inverse-U relationship

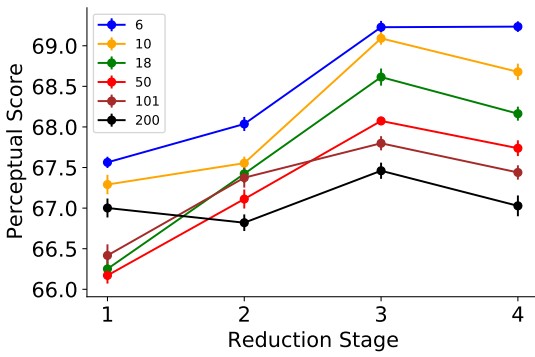 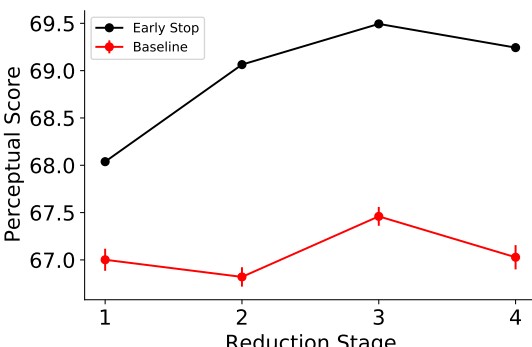

Figure 13: "Per-layer PS" for ResNets with different depths in Fig. 13a) and for a early-stopped ResNet-200 model with optimal PS in Fig. 13b.

Remember, PS is averaged over many layers, see Eq 2. However, it might be the case that optimal features for PS are buried in specific layers for better classifiers (e.g. lower layers), while other layers (e.g. high layers) exhibit different behaviour more optimal for classification. Therefore, we look at the best PS across layers.

We define the "layer-wise PS" $d(x_0^l, x_1^l)$ to be $\frac{1}{H_l W_l} \sum_{h,w} ||\tilde{y0}_{h,w}^l - \tilde{y1}_{h,w}^l||^2$. The PS in Eq 2 can be viewed as a mean ensemble of "layer-wise PS". The "optimal layer-wise PS" is then $\max_{l \in \mathcal{L}} d(x_0^l, x_1^l)$ Fig. 13a, displays the layer-wise PS for each of the 4 2D representations across different ResNet depths. The optimal $l$ for all depths is 3, and larger depths attain worse PS even at this optimal $l$. We additionally see that in Fig. 13b,

ResNet-200 under-performs the optimal layer-wise PS of its early-stopped variant at $l = 3$. **Therefore, we can conclude that sub-optimal features are not a cause of the inverse-U relationship.**

### 8.4 ImageNet class granularity cannot explain why ResNet-6 outperforms ResNet-200 on PS

ImageNet is a 1000 class classification problem that includes fine-grained classes. A classifier that models such classes successfully could have a reduced PS, since it could compromise on learning general features. The low accuracy of ResNet-6 implies that its capacity is sufficient to model only a subset of these classes, and its inability to model tougher classes might explain its high PS. We create random subsets having number of classes ranging from 50 to 900 and train ResNet-6 and ResNet-200 networks on each of these subsets. In Fig. 8.4, the PS gap between ResNet-200 and ResNet-6 reduces as the number of classes are decreased. But, ResNet-200 still under-performs ResNet-6. **Therefore, class granularity cannot fully explain why a less-accurate ResNet-6 significantly outperforms ResNet-200 on PS.** In Appendix E, we show similar results with a class subset selection strategy guided by a pretrained ResNet-6.

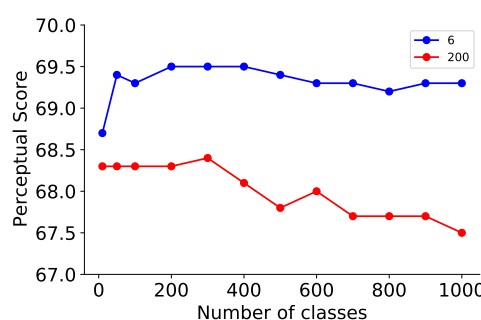

Figure 14: We create ImageNet subsets by randomly sampling $x$ number of classes and then train models on each of these subsets. The plot shows PS of ResNet-6 and ResNet-200 models vs number of ImageNet classes ($x$) in these subsets.

### 8.5 High PS features do not necessarily have high entropy

Wang et al. (2021a) show that ResNets are not suitable for style transfer due to the presence of skip connections. Skip connections result in features with low entropy which prevent capturing all style modes from a ground-truth style image. We explore if the mean entropy of activations can explain the inverse-U phenomenon between PS and accuracy. As done in Wang et al. (2021a), we convert intermediate activations $x \in \mathcal{R}^{H \times W \times C}$ into a probability distribution across $H \times W \times C$ values by applying a softmax transformation. We then report the average normalized entropy of this distribution across four 2-D representations on the BAPPS validation set.

Fig. 15a plots the normalized entropy of each of the four ResNet-200 reduction stages (marked 1 - 4) across training. As seen in Wang et al. (2021a), representations closer to the output at later reduction stages have a much lower entropy than representations closer to the input at earlier reduction stages. The entropy also decreases as a function of train steps. For the first few training epochs, where the entropy is between 0.9 and 1.0, there is a negative correlation between PS and mean activation entropy. Note that the maximum entropy is at initialization and not after a few training epochs where ResNet-200 obtains its highest PS. After the first few epochs, there is a positive correlation where entropy and PS both decrease during training. Fig 15a suggests that there is a optimal entropy as a function of train steps, where the PS peaks.

However, Fig. 15d shows that there is almost no correlation between entropy and PS across ResNets with various depths. ResNet-6 achieves the highest PS at a mean entropy of $\approx 0.5$ while ResNet-50 features have the highest entropy around 0.7 and have a much lower PS of 68.0. Therefore, entropy does not fully explain the observed effect.

We remove all skip connections in the ResNets to increase the entropy of the intermediate features as done in Wang et al. (2021a). Note that the maximum depth that we are able to successfully train without skip connections is 50. In Fig. 16a, removing skip connections increase the entropy across all depths. Specifically, ResNet-50 has a huge increase in entropy, making the activations close to a uniform distribution. Even after removing the skip connections in Fig. 16b and Fig. 16c, ResNet-6 and early-stopped ResNets attain the highest PS respectively similar to the baseline ResNets. While increasing the entropy of intermediate features can improve results of ResNets on style transfer, they don't improve PS.

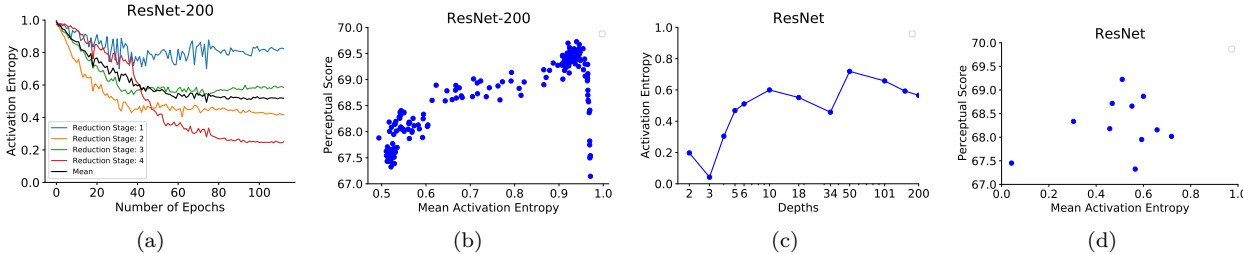

Figure 15: Fig. 15a depicts the entropy of each of the four ResNet reduction stages as a function of train epochs. The black line is the mean activation entropy across the 4 reduction stages. Fig. 15c plots the entropy as a function of depths in ResNets. Figs. 15b and 15d are the corresponding scatter plots between entropy and PS.

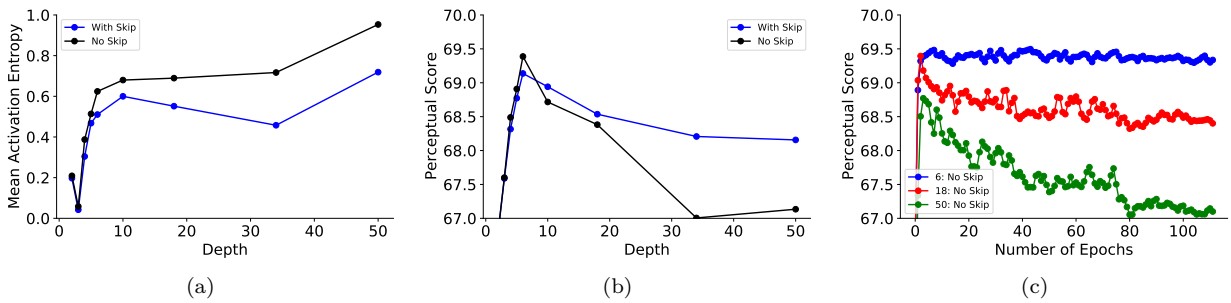

Figure 16: Figs. 16a shows the entropy of ResNet models **without skip connections** compared to baseline ResNets across all depths. Figs. 16b and 16c display the PS of **ResNets without skip connections** across depths and train epochs; the pattern is the same for ResNets with skip connections.

### 8.6 Low PS models are not necessarily more reliant on high frequency information for classification

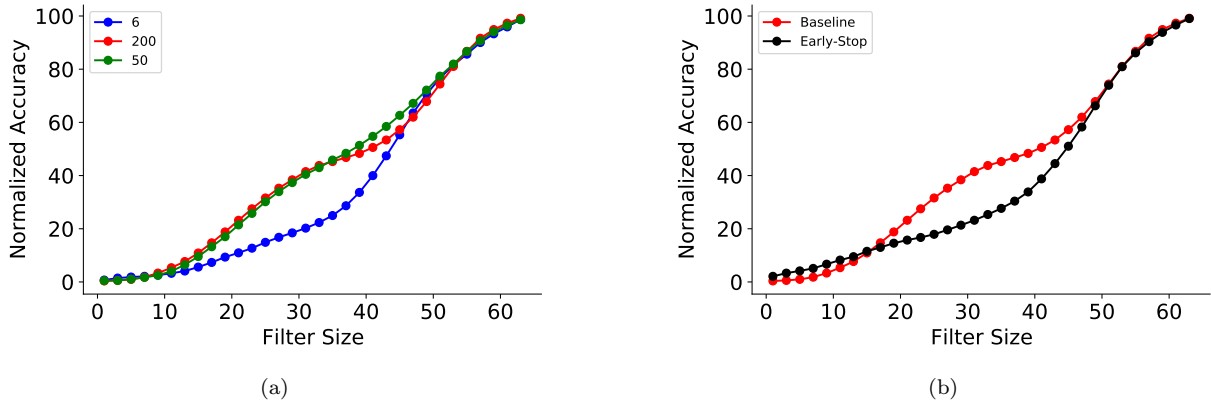

Figure 17: Normalized Accuracy as a function of low-pass filter sizes for ResNets trained with default hyperparameters. (Fig. 17a) and ResNet-200 - Baseline (converged) vs early-stopped (Fig. 17b)

Networks that rely more on high-frequency information for classification could be less robust to high-frequency distortions or removal of high frequencies from an image (Yin et al., 2019), and as an effect have low PS. We analyze the relationship between spatial frequency sensitivity of different networks and their PS. A low-pass square filter of side $r$ filters out the high frequencies in an image outside a square with

edge length $r$ in its Fourier spectrum. We measure the "normalized accuracy", which is the accuracy on low-pass filtered images divided by its accuracy on clean images as a function of $r$. A model more reliant on high frequency information will have a higher "normalized accuracy" slope at high values of $r$. ResNet-6 has a higher "normalized accuracy" slope at a high $r = 40$ to $50$ as compared to ResNet-6 (Fig. 17b). Despite being more reliant on higher spatial frequencies, ResNet-6 achieves a higher PS compared to ResNet-200. Similarly, ResNet 200 becomes more reliant on higher spatial frequencies if it is early stopped (Fig. 17a), while also increasing its PS (Fig. 5b). **These results indicate that models that have low PS are not necessarily more reliant on high frequency information for classification.**

## 9    Conclusion

In this paper, we explore the question if better classifiers can serve as better feature extractors for perceptual metrics. To answer this question, we conduct experiments across ResNets and ViTs across many different hyperparameters. Except for label smoothing and dropout, we see that PS exhibits an inverse-U relationship with accuracy across the hyperparameters we considered. We then probe a number of explanations for the inverse-U relationship involving skip connections, Global Similarity Functions, Distortion Sensitivity, Layer-wise Perceptual Scores, Spatial Frequency, Sensitivity, and ImageNet Class Granularity. While none of these explanations can offer an explanation for the observed tradeoff between ImageNet accuracy and perceptual similarity, we hope our paper opens the door for further research in this area.

**Broader Impact Statement**   Our results are based on BAPPS, which consists of exclusively low-level distortions as opposed to high-level semantic differences. We believe low-level distortions such as gaussian blur and color distortions are less likely to be susceptible to bias across different human categories as compared to high-level semantic features such as facial features. It is an open and interesting question whether different categories of humans like race and gender perceive low-level distortions differently. Increasing the diversity of both distortions and human labels in future perceptual similarity datasets is another interesting direction that might help to mitigate human biases.

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

## A   Perceptual Scores: High Resolution Models

We train out-of-the-box ImageNet models on their typical resolutions ($224 \times 224$) and above. We then resize the $64 \times 64$ BAPPS images to the resolutions on which the models are trained on and report their accuracies and PS in Figure 18a. The inverse-U relationship still exists. But as expected, the perceptual scores across all models are considerably worse. Our best and worst PS are 67.6 and 64.2 as compared to 69.1 and 67.3 obtained with models trained directly on $64 \times 64$ images. Therefore, we advise to train models on smaller images ($64 \times 64$) to have a high PS.

We also evaluate the PS of out-of-the-box ResNets trained on high-resolution images using $64 \times 64$ BAPPS images directly and observe the same phenomenon in 18b.

## B   Perceptual Score: Sub-dataset Breakdown

BAPPS consists of two sub-datasets "Distortions" and "Real Algorithms" that differ in the generative process of the target patches. In Figs. 21 and 22, we show scatter plots for accuracy-PS and PS vs hyper-parameter values for the "Distortion" subset. We show similar results for the "Real Algorithms" subset in Figs. 23 and 24. The inverse-U relationship generalizes across each sub-dataset of BAPPS.

## C   Perceptual Score: Learned

Previous work found that it is possible to improve the prediction of human binary choice on a dataset of bird images Attarian et al. (2020) and human 2-AFC judgements Zhang et al. (2018) by finding appropriate linear transformations. Through our paper is focused on assessing the inherent properties of ImageNet pretrained models to capture perceptual similarity, yet following the linear evaluation protocol in Zhang et al. (2018), we train a linear layer on top of pretrained ImageNet features to match supervised human judgements on BAPPS. The PS gap between ResNet-6 and ResNet-200 narrows down from 1.5 to 0.8, but even after training, there is a negative corelation among high-performing state-of-the-art residual networks in Fig. 19

## D   Perceptual Score: Learning Rate

We demonstrate that the inverse-U phenomenon also exists as a function of the peak learning rate in ResNets in 20.

## E   Class Granularity: More Results

We replace the "random sampling" strategy in our previous "Class Granularity" experiment with two variants. We compute ResNet-6's accuracy on each ImageNet class and rank the classes according to the per-class accuracy. In our "top" and "bottom" experiments, each subset of $k$ classes consists of the top $k$ and bottom $k$ classes according to this ranking respectively. In Figs. 25b and 25a, we vary $k$ from 50 to 900. As observed in our previous experiment, as the number of classes are reduced, ResNet-200's PS improves but still underperforms ResNet-6. Interestingly, the particular sampling strategy used does not seem to have a significant effect.

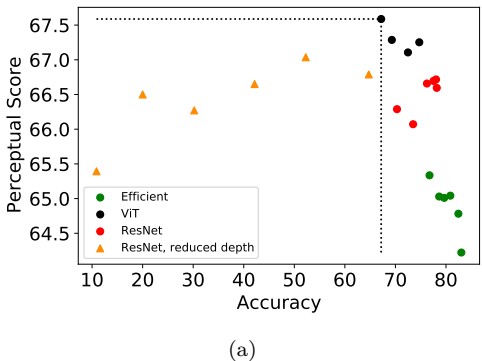
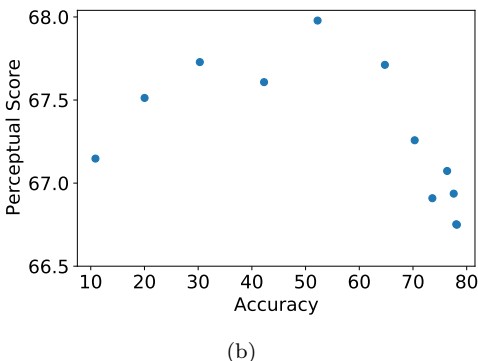

(a)

(b)

Figure 18: We train all out-of-the-box networks on their typical higher resolutions (224 × 224) and above. In Fig. 18a, we resize the 64 × 64 BAPPS images to match the resolutions at which the corresponding networks are trained. In Fig. 18b, we display the PS evaluated on 64 × 64 images directly on out-of-the-boz ResNets. The inverse-U relationship exists but the perceptual scores across all architectures are considerably worse

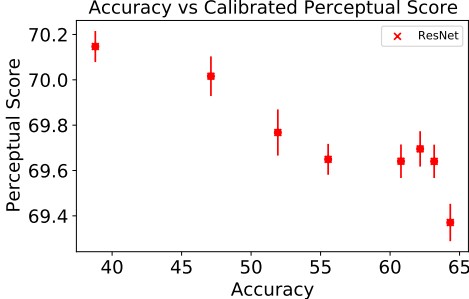

Figure 19: We train a linear layer on top of ImageNet features. The negative correlation exists among state-of-the-art residual networks.

## F  Spatial Frequency Analysis: Further Exploration

**Normalized Frequency Slope.**  Figs. 26a and 26b, we display the "normalized accuracy slope" as a function of $r$. The "normalized accuracy slope" at radius $r$ is the difference between the "normalized accuracy" at radius $r$ and the "normalized accuracy" at radius $r - 1$. It helps us understand: at which frequencies do different models have their highest accuracy gains? Models that have high PS such as ResNet-6 and early-stopped ResNet-200 have higher slopes at higher frequencies as compared to low PS models.

## G  Augmentations:

We investigate the effect of Gaussian Noise Augmentation and AutoAugment on PS. By construction, any augmentation strategy forces the network to produce the same outputs at the last layer of the network, for different distortions of an image. One can thus expect augmentation to make a network less sensitive to various distortions of an image, and as a result lower its PS.

**Gaussian Noise.**  For a given noise factor $s$, we first sample $\sigma \sim U(0, s)$ per-image. We then sample per-pixel noise independently from the Gaussian distribution $\mathcal{N}(0, \sigma)$, truncate it between 0.0 and 1.0 and add it to the original image. We show results for 4 noise factors, 0.1, 0.2, 0.5 and 1.0. The "Distortions" subset of BAPPS consists of Gaussian-noise based distortions, and Gaussian noise augmentation would make

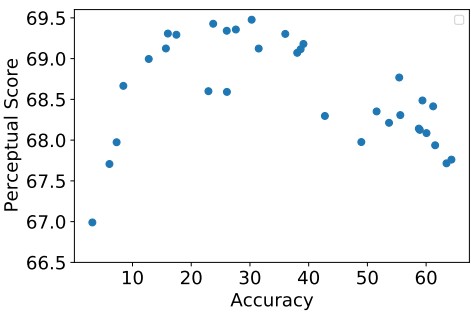

Figure 20: Inverse-U as a function of peak learning rates.

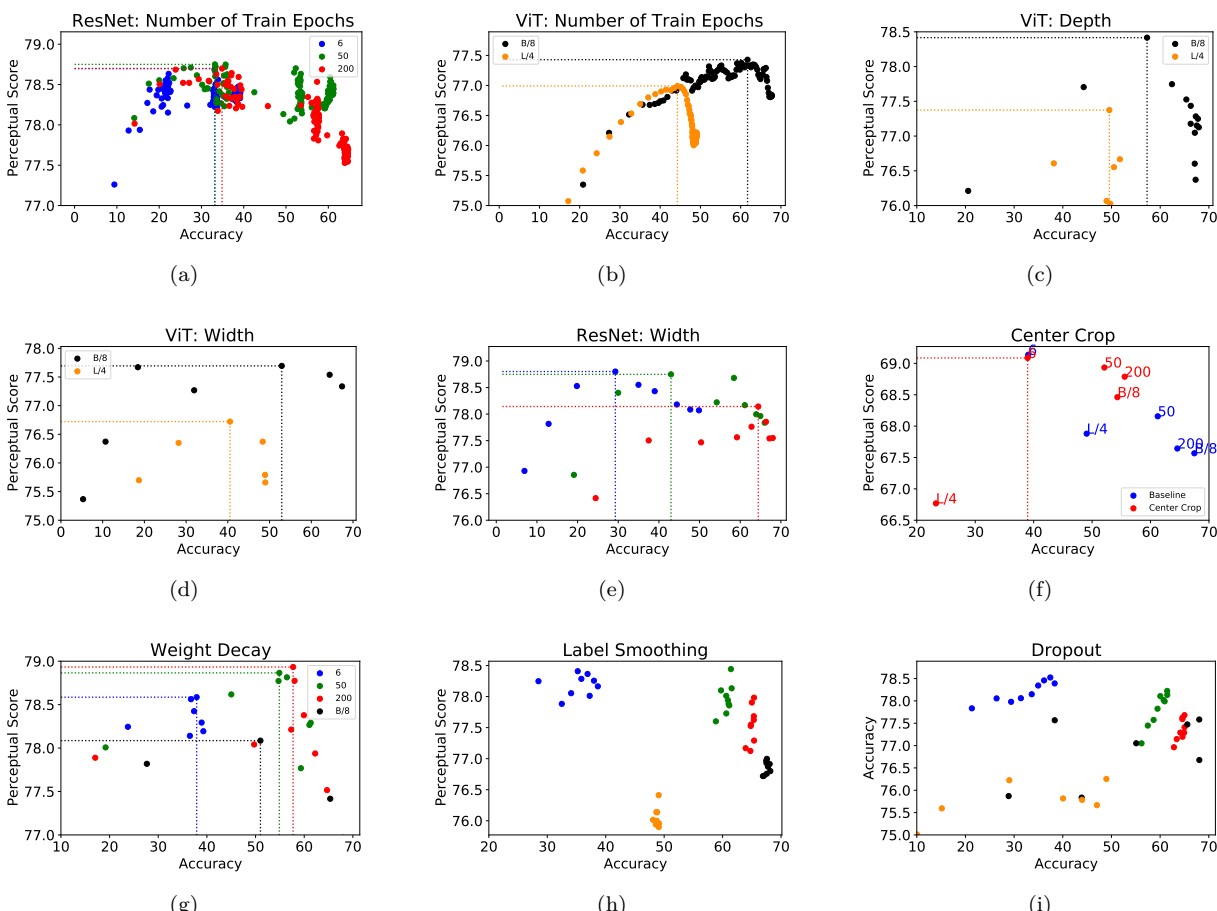

Figure 21: **Distortions:** The relationship between Perceptual Scores and accuracy when varying different hyperparameters on the **tradional algorithms** subset of BAPPS. Each plot depicts the ImageNet accuracy (x axis) and Perceptual Score (y axis) obtained by a 1D sweep over that particular hyperparameter.

the network less-sensitive to such distortions. As an effect, in Fig. 27b, an the noise factor increases, the PS on the "Distortions" subset of BAPPS decreases by a huge amount.

**AutoAugment.** We train models with a state-of-the-art augmentation technique AutoAugment Cubuk et al. (2019) that consists of a mixture of different augmentations. As seen in Figs. 27d, 27f, 27e, for every architecture, AutoAugment decreases both the accuracy and PS. We note that AutoAugment strategy was

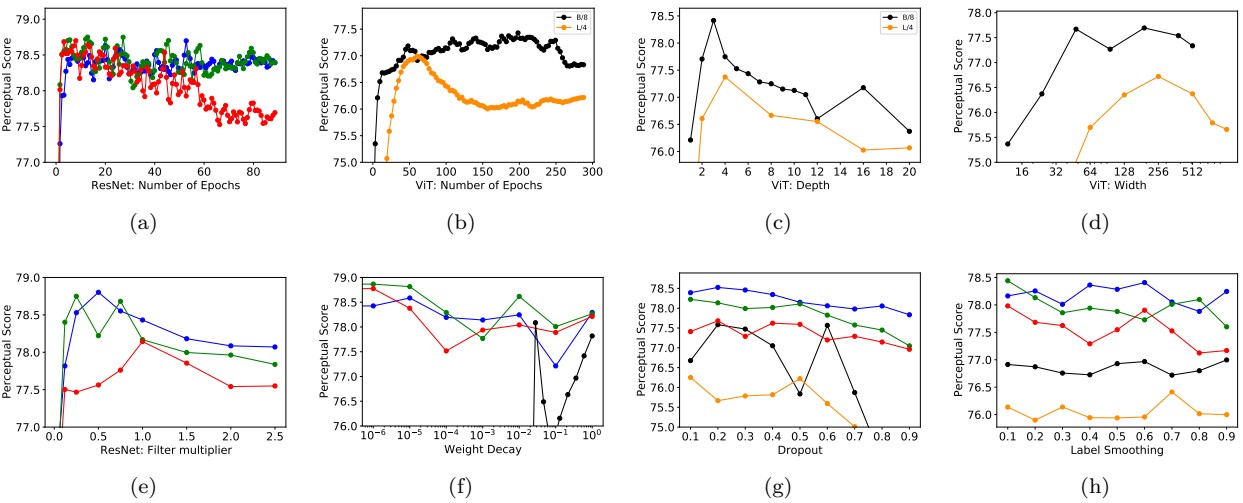

Figure 22: **Distortions:** In each plot, we vary a single hyperparameter along a 1D grid and plot the Perceptual Scores on the **distortions** subset of BAPPS

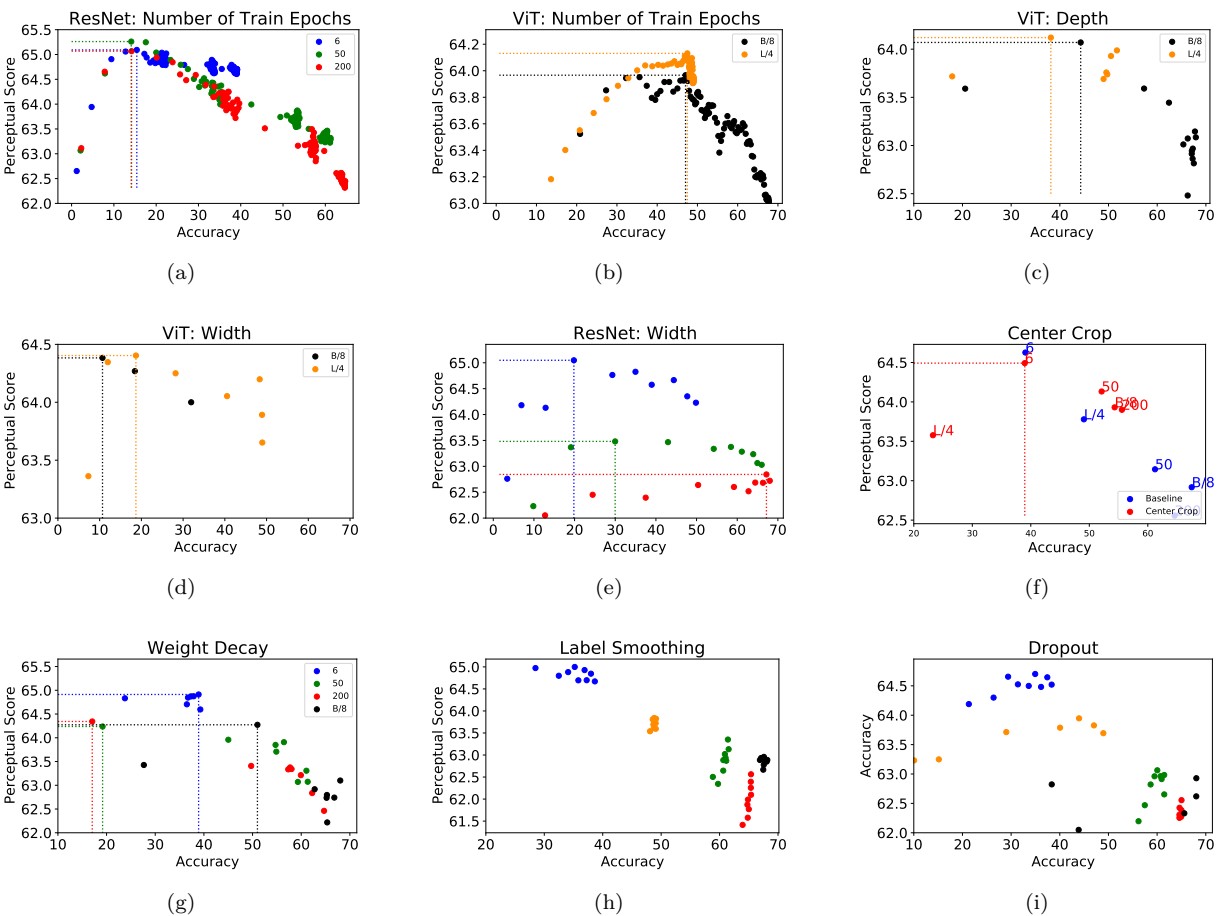

Figure 23: **Real Algorithms:** The relationship between Perceptual Scores and accuracy when varying different hyperparameters on the **real algorithms** subset of BAPPS. Each plot depicts the ImageNet accuracy (x axis) and Perceptual Score (y axis) obtained by a 1D sweep over that particular hyperparameter.

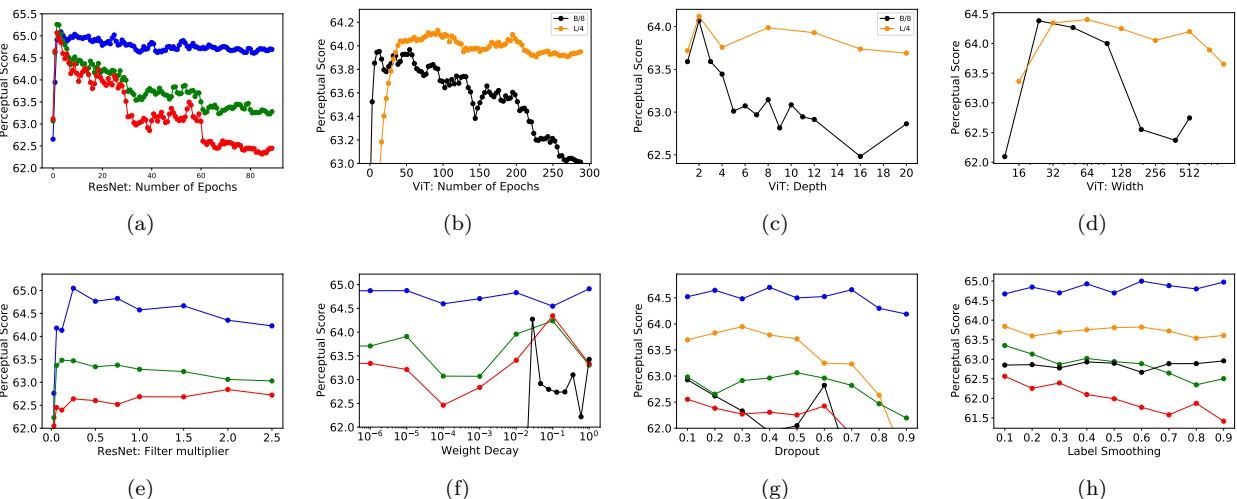

Figure 24: **Real Algorithms:** In each plot, we vary a single hyperparameter along a 1D grid and plot the Perceptual Scores on the **real algorithms** subset of BAPPS.

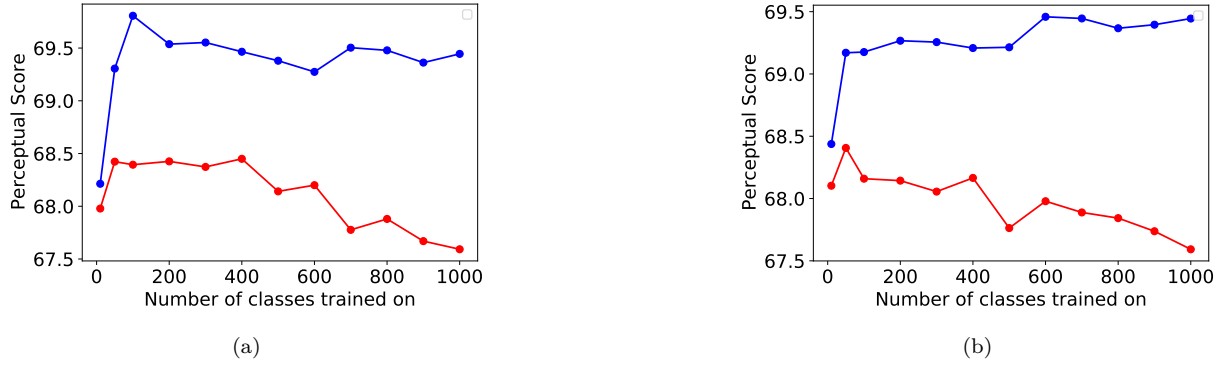

Figure 25: We dive deep into the impact of class granularity. We create class subsets according to ResNet-6's accuracy, instead of a random subset. In Fig. 25a and Fig. 25b, we show the accuracy and PS of the networks trained on the top and botton $x$ number of classes as given by ResNet-6's accuracy.

developed explicitly by maximizing the validation accuracy on high resolution ($224 \times 224$) images, and thus the policy might not transfer to low resolution ($64 \times 64$) images. This can explain the decrease in accuracy

## H  Accuracy vs Hyperparameters

We plot the accuracy as a function of each hyperparameter where we observe the inverse-U relationshop between validation accuracy and Perceptual Score in Fig. 28. We note that the validation accuracy steadily increases as a function of the hyperparameter. The peak in Perceptual Scores occur in the underfitting regime where the ImageNet models have poor to moderate accuracies.

## I  Efficient Net: Perceptual Score

In Fig.29, we observe the inverse-U tradeoff in EfficientNet-B0 and EfficientNet-B5 on varying function of depth, width and number of epochs respectively. EfficientNets attain their peak PS at extremely small width, depth and number of train steps.

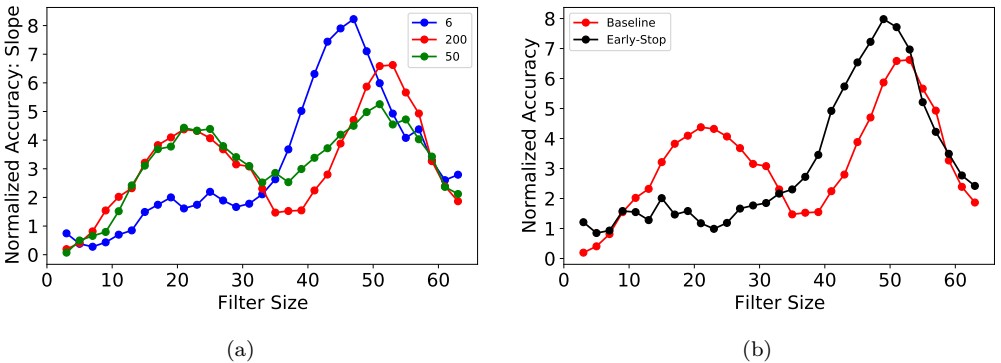

Figure 26: We further explore the spatial frequency responses to networks via **Normalized Frequency Slopes:** (Figs. 26a and 26b)

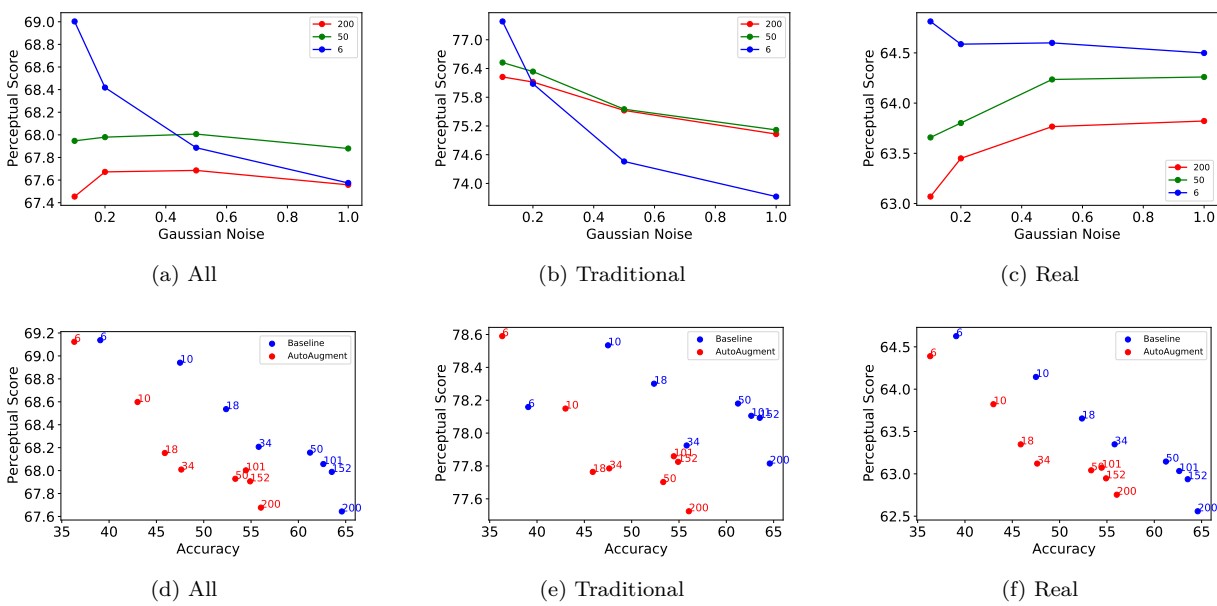

Figure 27: We investigate the effect of Gaussian Noise Augmentation and AutoAugment strategies on PS

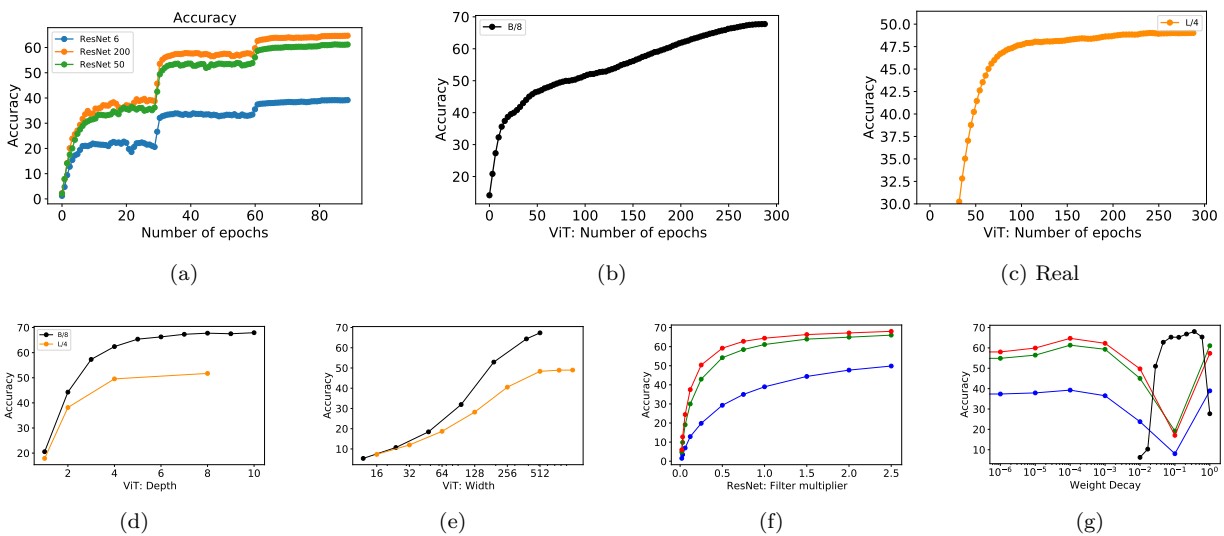

Figure 28: In each plot, we vary a single hyperparameter along a 1D grid and plot the validation accuracy.

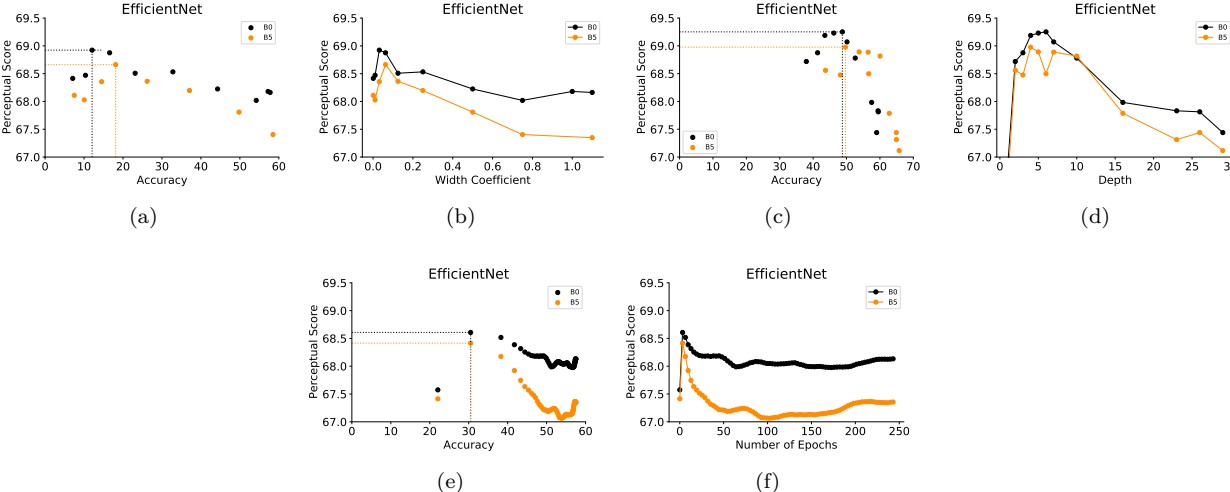

Figure 29: Figs. 29a, 29c and 29e show PS vs accuracy on varying width, depth and number of epochs in EfficientNets. Figs. 29b, 29d and 29f depict PS as a function of width, depth and number of epochs in EfficientNets.

## J Distance Margins: Rank Correlation

The scoring function (Eq. 2) takes into account only if $d_1 > d_0$, and not the actual margins $(d_1 - d_0)$ themselves. To assess if the inverse-U relationship is agnostic to this choice, we use a scoring function that takes the distance margins into account. Namely, we measure the rank correlation between the distance margins between the distance margins $(d_1 - d_0)$ and the ground truth probabilities $p$ in BAPPS. A model having a high rank correlation, should assign large positive distance margins at high probabilities and large negative distance margins at low probabilities. In Fig. 30, we show that similar to PS, rank correlation exhibits an inverse-U relationship for a) ResNets at different depths b) ResNet-200 as a function of train epochs. Therefore the conclusion remains unchanged.

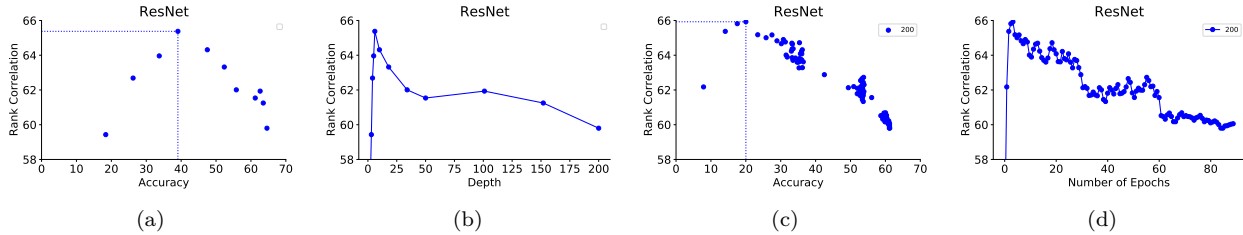

Figure 30: We show that the rank correlation between the distance margins $d_1 - d_0$ in 2 and $p$ in BAPPS exhibits an inverse-U behaviour. Figs. 30a and 30c show the rank correlation vs accuracy on varying ResNet depths and train epochs. Figs. 30b and 30d depict the rank correlation as a function of ResNet depths and train epochs.

## K    Inverse-U on TID2013

BAPPS was explicitly constructed using $64 \times 64$ images so that human raters can focus on low-level local changes as opposed to high-level semantic differences. We show results on the TID2013 dataset (Ponomarenko et al., 2015), which consists of higher resolution images $384 \times 512$. Each example in TID2013 consists of a reference image $x$, a distorted image $y$ and a mean opinion score rating $s$. For every model, we evaluate the rank correlation between $s$ and $d(x, y)$ (as given in Eq. 1). Firstly, we assess ResNets of various depths trained on ImageNet $224 \times 224$. Secondly, we estimate the rank correlation of a ResNet-50 model trained on ImageNet $384 \times 384$ during the course of training. Fig. 31b and 31d show that shallow ResNets and early-stopped ResNets achieve the highest rank correlation. In Figs. 31a and 31c, we see that the inverse-U relationship exists on TID2013.

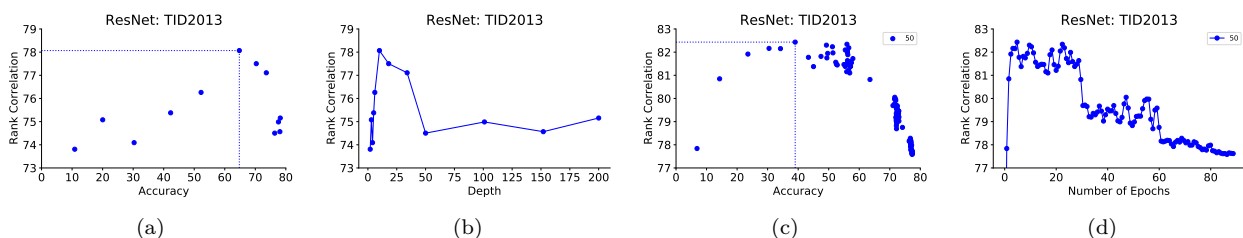

Figure 31: Figs. 31a and 31c show the rank correlation vs accuracy on varying the depth and train epochs on TID2013 ($384 \times 512$). Figs. 31b and 31d depict the rank correlation vs ResNet depths and train epochs on TID2013 ($384 \times 512$)

## L    Code Snippets

We present code snippets for the different distance functions used in our paper.

Listing 1: Code Snippets for different perceptual functions

```
def perceptual(tensor1, tensor2, eps=1e-10):
    """Default perceptual distance function.

    Args:
        tensor1: shape=(B, H, W, C)
        tensor2: shape=(B, H, W, C)
    Returns:
        dist: shape=(B,)
```

```python
    """
    tensor1_n = np.linalg.norm(tensor1, ord=2, axis=-1, keepdims=True)
    tensor2_n = np.linalg.norm(tensor2, ord=2, axis=-1, keepdims=True)
    tensor1 = tensor1 / (tensor1_n + eps)
    tensor2 = tensor2 / (tensor2_n + eps)
    dist = np.sum((tensor1 - tensor2)**2, axis=-1)
    dist = np.mean(dist, axis=(1, 2))
    return dist

def mean_pool(tensor1, tensor2, eps=1e-10):
    """Mean Pool perceptual distance function.

    Args:
        tensor1: shape=(B, H, W, C)
        tensor2: shape=(B, H, W, C)
    Returns:
        dist: shape=(B,)
    """
    tensor1 = np.mean(tensor1, (1, 2), keepdims=True)
    tensor2 = np.mean(tensor2, (1, 2), keepdims=True)
    return perceptual(tensor1, tensor2, eps=eps)

def compute_gram(tensor, eps):
    """Returns (BxCxC) cross correlation."""
    _, h, w, c = tensor.shape
    tensor = np.reshape(tensor, (-1, h*w, c))
    tensor_norm = np.linalg.norm(tensor, ord=2, axis=1, keepdims=True)
    tensor = tensor / (tensor_norm + eps)
    # Channel-wise cross correlation.
    tensor_t = np.transpose(tensor, (0, 2, 1))
    return np.matmul(tensor_t, tensor)

def style(tensor1, tensor2, eps=1e-10):
    """Style perceptual distance function.

    Args:
        tensor1: shape=(B, H, W, C)
        tensor2: shape=(B, H, W, C)
    Returns:
        dist: shape=(B,)
    """
    tensor1_gram = compute_gram(tensor1, eps=eps)
    tensor2_gram = compute_gram(tensor2, eps=eps)
    dist = tensor1_gram - tensor2_gram
    return np.mean(dist**2, axis=(1, 2))
```

## M  Default Hyper-parameters

We provide the default training hyper-parameters for the ResNets, EfficientNets and Vision Transformers in Tables 2, 3, 4 and 5.

| Hyper-Parameter | Value |
| --- | --- |
| Batch Size | 1024 |
| Base Learning Rate | 0.1 |
| Train Steps | 112590 |
| Momentum | 0.9 |
| Weight Decay | 0.0001 |
| Label Smoothing | 0.0 |
| LR Schedule | Step-wise Decay |
| Batch-Norm Momentum | 0.9 |

Table 2: ResNet: Default Hyperparameters

| Hyper-Parameter | Value |
| --- | --- |
| Batch Size | 2048 |
| Base Learning Rate | 0.128 |
| Train Steps | 218949 |
| Optimizer | RMSProp |
| Momentum | 0.9 |
| Weight Decay | 1e-5 |
| Label Smoothing | 0.1 |
| LR Schedule | Warmup + Exp Decay |
| Batch-Norm Momentum | 0.9 |
| Polyak Average | 0.9999 |

Table 3: EfficientNet: Default Hyperparameters

| Hyper-Parameter | Value |
| --- | --- |
| Batch Size | 4096 |
| Base Learning Rate | 3e-3 |
| LR Schedule | Warmup + Cosine |
| LR Warmup Steps | 10000 |
| Train Steps | 93834 |
| Optimizer | Adam |
| Beta1 | 0.9 |
| Beta2 | 0.999 |
| Weight Decay | 0.3 |
| Hidden Size | 768 |
| MLP Dim | 3072 |
| Number of Layers | 12 |
| Number of Heads | 12 |
| Dropout | 0.1 |
| Label Smoothing | 0.1 |

Table 4: ViT-B/8: Default Hyperparameters

| Hyper-Parameter | Value |
| --- | --- |
| Batch Size | 4096 |
| Base Learning Rate | 1e-4 |
| LR Schedule | Warmup + Cosine |
| LR Warmup Steps | 10000 |
| Train Steps | 93834 |
| Optimizer | Adam |
| Beta1 | 0.9 |
| Beta2 | 0.999 |
| Weight Decay | 0.01 |
| Hidden Size | 1024 |
| MLP Dim | 4096 |
| Number of Layers | 24 |
| Number of Heads | 12 |
| Dropout | 0.1 |
| Label Smoothing | 0.1 |

Table 5: ViT-L/4: Default Hyperparameters

