# OpenReview forum: "Do better ImageNet classifiers assess perceptual similarity better?"
_TMLR — Accepted by TMLR_

### Review · Reviewer_7meX · 2022-07-11

**Summary Of Contributions:**

The authors study the perceptual similarity metric for different models. So far, the community has mostly used the AlexNet or VGG features to calculate the perceptual similarity metric. In this paper, the authors perform a wide range of experiments to study other possible feature extractors for this metric. They find an inverse V relationship between accuracy and perceptual similarity.

**Broader Impact Concerns:**

I do not have any concerns in regards to ethical implications of this work.

**Requested Changes:**

### Questions to authors:
- As far as I know, researchers still use pretrained VGG features for style transfer and ResNet models do not work as well as VGG. Is this a similar phenomenon as the one observed in this paper? In this paper (https://arxiv.org/abs/2104.05623), the authors study the reasons why ResNets perform worse than VGG for style transfer, and find that the issue lies in the skip connections which lead to low entropy features. Could it be that the same issue happens here? Entropy is decreased over the course of training after the network has found some optimum it starts converging into, and this may explain the decreased PS scores?
- The authors have only tested models with skip connections. Do VGG and AlexNet have a similar accuracy/PS score curve compared to the ResNets/ViTs over training? Please test these models to study the impact of skip connections on the observed phenomenon.
- Please go through my detailed review and incorporate the suggestions.


**Strengths And Weaknesses:**

### Strenghts:
- The related work section is well researched (as far as I can judge) and well written. The authors include the relevant papers and position their work well in this context.
- I really liked the paper until the results section which I found quite confusing. But I think the dissemination of the results should be fixable, and then it would be a really nice paper.
- The paper shows a high level of rigor in the conducted experiments which is great.
- I believe the research question is interesting to the computer vision research community.

### Weaknesses:
- I have written a very detailed review below which details all issues I have with this submission.
- I think most of my points are well addressable, and are mostly issues with the presentation of the results/ some formulations/ typos. I strongly advise the authors to restructure their results section as it is very hard to parse currently, and impossible to follow in some occasions.
- I would restructure the results such that the subsection title of the results says e.g. "The correlation between perceptual similarity and accuracy follows an inverse V-shape for a wide range of models over the course of training". Then show the Figs 3a and 3b below the paragraph. This will make it much easier for the reader to parse the results.
- The BAPPS dataset should be explained better and example images of the task should be provided.
- A summary and a discussion of the findings at the end of the paper would be nice. Currently, the authors present their main findings in Figure 1 and conduct ablations to find out the possible causes. But they never summarize which of the causes they think contributes most/ have any form of a more high-level discussion of their results. This is especially a problem, because the authors show SO many results, such that the reader simply gets lost. I have linked a paper in the “Requested Changes” field which I think may explain their findings and think the authors should look into that paper and see whether the same argument applies in this case. As a summary of this point, I currently understand that there is a pareto frontier between PS and accuracy, but I have no idea why.


### General comments:
Eq. 2: It would be nice to give some intuition on eq. 2. I started inserting random numbers for d1, d2 and p to try to understand how it behaves. It would be helpful to draw a graph with synthetic data for some combinations of (d1,d2,p), such as e.g., for typical values of p and varying d1 and d2. This will just give the reader a better intuition how this PS score behaves. This is crucial as this quantity seems to be central for understanding the paper. The quantities from the “Dynamic Range” paragraph could be added into this graph as lines. Currently, the paragraph “Dynamic Range” is hard to grasp /visualize.

Page 4: “We train our networks on ImageNet at a resolution of 64 × 64 and report their accuracies on the ImageNet validation set and PS on BAPPS.” The argumentation around using 64x64 images makes sense since this is also used in BAPPS. Could the authors comment on whether they think their results would generalize to the larger 224x224 resolution which is commonly used?

Page 4: When introducing the BAPPS dataset, please show example images and provide more details what this dataset shows. Are there near-duplicates which the PS score is supposed to find? Are there style similarities or corrupted inputs which the PS score should track? Please show examples for the reference image x and the target images. Are there style-content conflicts? If the query image is a sketch of a dog and the target images are 1) an image of a real dog and 2) a sketch of a cat, which do we think would be more similar in terms of what human subjects say, and what does the PS metric say is more similar? Given that the used PS metric correlates strongly with the “style” PS metric (Figs 6a+6b), I think the similarity between the query image and the target image (2) would be picked.

Page 5: Paragraph “representations”: Please specify which layers have been used to calculate the PS score in the literature here. I assume that the last sentence (“Our initial experiments on using features at the input of every layer instead of every block or reduction stage lead to worse PS across all architectures.”) addresses this point. Please make it more specific.

Page 5: “All modern networks (red, green, and black) in Fig. 2 obtain a higher PS than FSIMc and randomly initialized networks.” Please add the FSIMc and the randomly initializeds networks PS numbers in brackets here.

Page 6: “ResNet-6 happens to achieve this optimal accuracy with a PS of 69.1, outperforming AlexNet.” Please add the Alexnet number here.

Page 6: “3) ResNet-200 and ResNet-3 achieve similar PS, while having a 45% difference in accuracy.” I believe one should say “a difference of 45 percentage points”.


Fig 3 is ordered quite chaotically. I would group the ResNets and ViTs in the first 5 graphs and not interchange them all the time. Legends are missing in some graphs. Please add legends in all graphs as I find the graphs currently hard to read, because the legends change from graph to graph.

Fig 4: legend for the resnets is missing.

[IMPORTANT] Fig 3 + Fig 4: Please use the same y-axis in all plots, at least in those with the same architectures. Then, one will be able to also judge how strongly the different hyperparameters influence the PS scores. This is true for all Figures, e.g. 6a+6b.

Section 8 is very confusing in terms of writing and presentation of results. First, Figs. 6a and 6b should be separate Figures and not included in the 9x9 figures plot. I think it would be much better if the authors could break this section into parts where they write one question they want to analyze, e.g. comparing the style / mean pool loss to the local loss and then showing the Figures 6a+6b directly underneath the text with the discussion. Similarly, the current Figs 6c+6d should be placed separately below the distortion sensitivity subsection.

I would advise the authors to rename their subsection headings in section 8 (and possibly the other results sections as well) such that the result is evident from the title, e.g., 8.2. could say “Decreased distortion sensitivity of stronger models does not explain their lower PS score”. This will make it easier to understand the results and is a quite customary way of disseminating results.

“In Figs. 6a and 6b, we present scatter plots between accuracy and PS with the style and mean pool similarity functions.” What do 6a and 6b show? In terms of notation, they look exactly the same to me. Looking at the caption, I see that “(Fig. 6a: Out-of-the-box Networks and Fig. 6b: ResNet-
200 on varying Train Epochs)” What does “out-of-the-box networks” mean?


“ResNet-6 with its Mean Pool and Style variants outperform the baseline (local) 69.1 with scores of 69.7 and 69.5 respectively. On early-stopped ResNet-200, the mean pool and style functions improve upon the baseline score of 69.5 with 69.8 and 69.7 respectively. Early-stopped ResNet-6 with its mean pool variant achieves 70.2.” ResNet6 is referenced twice with different numbers. Does the text relate to Fig.6a or 6b or some other plot?


Page 10: “The PS gap between ResNet-6 and ResNet-200 narrows down from 1.5 to 0.8, but even after
training, the ResNet-6 still outperforms the ResNet-200. See the appendix for more details.” Please add more detailed references to the Appendix, e.g. “see appendix A.12 for more details.”


Page 10: “Intuitively, better networks will be less sensitive to the distortions introduced in the BAPPS
dataset, since the class will not change under these distortions.” You should describe the BAPPS dataset in detail earlier in the paper, such that the reader knows that there are distortions in this dataset.


Page 10: “Learned.” Very strange heading for a paragraph.

Page 11: “Let xf and xn be the farther and nearer patch to a reference image x, respectively.” What are farther or nearer patches? Please provide example images, I currently have no intuition what you mean here.

It says Fig 6e twice. I believe it should be 6i.

If the authors want the reader to read through the appendix, they should refer to it in the paper. Otherwise, the reader has no incentives to open the appendix.

### Minor:

Fig. 1 is not referenced in the text in the introduction. Fig. 1 is referenced later, but should be also mentioned in the intro since it occurs so early in the paper.

Page 3: „Hermann et al. Hermann et al. (2019) show that among high-performing models, shape-bias is correlated with ImageNet accuracy.” Should be \citep{} not \citet{} here.

Page 4: “Unlike accuracy that can vary from 0 to 100, PS have a narrow dynamic range.” PS has …

Page 5: “Additionally, all networks perform worse than A lexNet.” Typo

Page 6: “In particular, we assess this relationship between accuracy and PS as a function of via 1D
hyperparameter sweeps.” Typo

Page 6: “2) PS against the hyperparameter values ((Fig. 4).” Typo


Fig. 3 + Fig4: I would remove the top+right canvas axes as described in: https://stackoverflow.com/questions/925024/how-can-i-remove-the-top-and-right-axis-in-matplotlib


Page 9: “we uncover a global Pareto frontier between PS and accuracies, see Fig.. 1.” Typo

---

> ### Author Response · Authors · 2022-07-26
> **Response (1 of N)**
>
> Thanks for the extremely helpful and detailed review. It has improved our work by quite a bit and we have incorporated all suggestions in the manuscript. All of our comments have been incorporated into the manuscript with the color **red**
>
> ---
>
> ### Models without skip connections
>
> We added **Section 8.4** exploring the relationship between the entropy of activations and PS. We additionally removed skip connections from ResNets and **show that the inverse-U persists**.  Here we describe the Section.
>
> [Wang et al 2021] show that ResNets are not suitable for style transfer because of skip connections. Skip connections result in features with low entropy, which in turn prevent capturing all style modes from the ground-truth style image. We now explore if features with lower entropy can explain the inverse-U phenomenon. As done in [Wang et al 2021], we convert $x \in \mathcal{R}^{H \times W \times C}$ into a probability distribution across $H \times W \times C$ values by applying a softmax transformation. We then report the average normalized entropy of this distribution across the 4 representations.
>
> Fig. 14 a plots the normalized entropy of each ResNet-200 representation across training. As seen in [Wang et al 2021] representations at later stages have a much lower entropy than earlier stages. The entropy also decreases as a function of train steps. However, the maximum entropy is at initialization and not after a few training epochs where ResNet-200 obtains its highest PS. Further, Fig. 14 d shows that there is almost no correlation between entropy and PS across ResNets with various depths. ResNet-6 achieves the highest PS at a mean entropy of $\approx 0.5$ while ResNet-50 features have the highest entropy around 0.7 and have a much lower PS of 68.0.
>
> We then remove all skip connections in the ResNet models to increase the entropy of the intermediate features as done in [Wang et al 2021]. Note that the maximum depth that we are able to successfully train is 50. In Fig. 15 a, removing the skip connections, increases the entropy across all depths and especially ResNet-50, which has an entropy close to 1.0. Even after removing the skip connections, we see in Fig. 15 b and Fig. 15c that ResNet-6 and early-stopped ResNets attain the highest PS similar to the baselines
>
> Note that the ResNets without skip connections are quite similar to VGG networks module the ordering of convolutions, max pooling layers and batch norm layers, which are shown not to have a significant effect on style transfer in [Wang et al 2021]. We have provided sufficient evidence that skip connections do not explain the inverse-U phenomenon.
>
> ----
> ### The BAPPS Dataset
>
> The BAPPS Dataset contains low-level distortions applied to images, which largely preserve semantics. The texture-shape conflict that you describe has been studied in [Hermann et al. 2020], albeit only in the final prediction layer. They show a positive correlation for shape bias among high performing ImageNet classifiers in Figure 4 of their paper. We added **Section 3.1 and Figure 2** for futher information on the BAPPS dataset and sample examples.
>
> The BAPPS Dataset is a dataset of 161k patches derived by applying exclusively low-level distortions to the MIT-Adobe 5k dataset for training and the RAISE1k dataset for validation. They consider 6 distortion families namely:
>
> * Traditional Distortions: Random noise, blurring, spatial shifts, corruptions and compression artifacts. (1 family)
> * CNN-based Distortions: Distortions created by CNN-based autoencoders trained on autoencoding, denoising, colorization and superresolution. (1 family)
> * Outputs of Real algorithms: Outputs from state-of-the-art frame interpolation, video deblurring, colorization and superresolution models. The distortions created by each class of models is treated as a separate family. (4 families)
>
> See Table 2 of [Zhang 2018 et al] for a comprehensive list of distortions. The train set consists of the traditional and CNN-based distortions and the validation set contains all 6 families. Given a family of distortions and a set of reference images, they generate the BAPPS dataset as follows. They select a reference patch $x$ and then apply two distortions at random to generate the target patches $x_0$ and $x_1$. They record the binary response of a human, indicating which of the target patches is closer to the reference patch. For a given image triplet, $(x_0, x_1, x)$, $p$ is the average of 2 and 5 human responses on the train and validation set respectively. Fig.2 displays 3 sample image triplets from the BAPPS Dataset.
>
> ----
>
> ### Do results generalize to the larger 224x224 resolution which is commonly used?
> In Appendix A, we presented results on models trained on high resolution (224x224) images. We observe similar phenomenon on both 64x64 BAPPS Images and 64x64 BAPPS resized to 224x224 with significantly worse PS. We now reference this directly in the **paragraph Training in Section 4.**

---

> ### Author Response · Authors · 2022-07-26
> **Response (2 of N)**
>
> ### PS Score: Dynamic Range
>
> As requested we simulated the upper and lower bounds of PS on synthetic data and added the results to **Section 3** with **Figure 3**
>
> To provide intuition on this dynamic range, we plot the mean PS for ground truth human ratings $p$ from BAPPS and simulated combinations of $(d_0, d_1)$ dependent on $p$. Let $\tilde{p} = \textbf{1} [p > 0.5]$, i.e we convert the human ratings into binary labels. If $\tilde{p} = 1$, we sample $d_0$ and $d_1$ from truncated normal distributions, i.e $d_0 \sim \mathcal{\phi}(1, \sigma, 0, 1), d_1 \sim \mathcal{\phi}(0, \sigma, 0, 1)$ where 0 and 1 are the lower and upper bounds. Conversely if $\tilde{p} = 0$, we sample $d_0 \sim \mathcal{\phi}(0, \sigma, 0, 1), d_1 \sim \mathcal{\phi}(1, \sigma, 0, 1)$. In Fig. 3, as $\sigma$ is increased, the distances $(d_0, d_1)$ have a higher chance of being misaligned with $p$, and as expected PS smoothly decreases from the machine upper bound to random choice.
>
>
> -----
>
> ### Representations
>
> We expanded the paragraph in **Section 4**
>
> In prior work, [Zhang et al] compute the PS of VGG using the outputs of every 2x2 max pooling layer, leading to 5 features in total. For the shallower AlexNet architecture, they employ the output of each of the 5 convolutions.
>
> Similar to VGG, for ResNets and EfficientNets, we use the outputs of the 4 reduction stages. …. Our initial experiments on using features at the output of every layer instead of every block or reduction stage, as computed in AlexNet} lead to worse PS across all architectures.
>
> -----
>
> ### Section 3
>
> * We moved the figures from **Figure 3** and **Figure 4** to just below their corresponding paragraphs.
> * ResNets and ViTs have different PS ranges. So, we adjusted the y-axis such that it is between 67.0 to 70.0 for ResNets in all plots and between 66.0 to 69.0 for ViT in all plots. We also decoupled the ResNet and ViT models in all plots.
>
> -----
>
> ### Section 8
>
> * **Show the subfigures of Section 8 directly under the corresponding texts**: Done. Each subfigure is now under the corresponding text. Additionally, we have expanded the captions of each subfigure.
> * **What do 6a and 6b show?** We expanded the caption and the subsection with the following text. Fig. 11 a displays the accuracy and PS of ResNets and EfficientNets trained with their default hyperparameters. Each point in 11b represents a ResNet-200 model at a different epoch during the course of training.
> * **ResNet6 is referenced twice with different numbers**
> The first reference is to the ResNet-6 model trained with the default hyperparameters and the second refers to an improved model which is early stopped (Table 1). We have made this explicit by adding the following text. In Fig 7a, ResNet-6 with its Mean Pool and Style variants outperform the baseline (local) 69.1 with scores of 69.7 and 69.5 respectively. In Fig. 7b, for a ResNet-200 model that is early-stopped, the mean pool and style functions improve upon the baseline score of 69.5 with 69.8 and 69.7 respectively. We additionally observe that the optimal early-stopped ResNet-6 from Table 1 further improves its performance from 69.7 with its mean pool variant achieving a PS of 70.2
>
> * **Learned**
> We renamed this paragraph from learned to learned linear layer on pretrained features. We also added the precise reference to the appendix.
> * **Distortion Sensitivity**
>   * We added a subsection about the distortions in BAPPS along with example images.
>   * **What are farther or nearer patches?**
> We reworded this section as follows to make the notation clearer. We also added a few images in Figure 2.
> From the BAPPS dataset, we retain only the examples, where the human raters unanimously agree that one of the target patches is closer to the reference patch than the other, i.e $p=1.0$ or $p=0.0$. For each such triplet ($x_0, x, x_1$) where $p=1.0$ or $p=0.0$, we denote $x_f$ to be the farther patch and $x_n$ to be the nearer patch. Concretely, in Eq 2, when $p=1.0, x_f=x_0, x_n=x_1$ or $p=0.0, x_f=x_1, x_n=x_0$.
> * **Subsection Renaming:** We renamed each subsection in 8 as following:
>   * Global Similarity Function -> The inverse-U phenomenon persists with improved perceptual similarity functions
>   * Distortion Sensitivity -> Low PS models are not necessarily less sensitive to distortions
>   * Layer-wise Perceptual Scores -> Sub-optimal features are not a cause of the inverse-U relationship
>   * Spatial Frequency Sensitivity -> Low PS models are not necessarily more reliant on high frequency information for classification.
>   * ImageNet class granularity:  ImageNet class granularity cannot explain why ResNet-6 outperforms ResNet-200 on PS
>
> ---

---

> ### Author Response · Authors · 2022-07-26
> **Response (3 of N)**
>
> ### Reference appendix in main paper:
>
> We reference the appendix now in the main section.
> * **Section 4**: Nonetheless in Appendix A, we observe similar phenomena on models trained with high resolution images, with significantly worse PS.
> * **Section 5**:  Appendix I contains a list of default hyperparameters.
> * **Section 6** In Appendix H, for all our sweeps, the validation accuracy does not decrease during training, indicating none of the networks overfit on the ImageNet train set
> * **Section 8.1** See Appendix C for more details under learned linear layers.
> * **Section 8.5** In Appendix E, we show similar results with a class subset selection strategy guided by a pretrained ResNet-6.
>
> ----
>
> ### Other
>
> * It says Fig 6e twice. I believe it should be 6i: Done and moved to a separate figure.
> * Add the FSIMc and the randomly initialized networks PS numbers here: Done
> * Please add the Alexnet number here: Done
> * A difference of 45 percentage points: Done
>
> ----
>
> ### Minor:
>
> * Fig 1, reference. We now reference Figure 1 in the 4th paragraph.
> * Page 3: Hermann et al. Hermann et al. (2019) Should be \citep{} not \citet{} here. Done
> * Page 4: have -> has. Done
> * Page 5: A lexNet -> AlexNet. Done.
> * Page 6: Changed to as a function of hyperparameters via 1-D sweeps.
> * Page 6: Typo -> Fixed.
> * Page 9: Fixed.
> * Remove Top-Right canvas: Done
>
> ----

---

> ### Author Response · Authors · 2022-07-26
> **Response: Final**
>
> The **changes since last submission**  section at the top contain a summary of the changes we made in response to your review. We believe we have addressed all your comments and concerns and look forward to your updated impression of our paper.

---

> > ### Comment · Reviewer_7meX · 2022-07-29
> > **Response to the authors' response**
> >
> > I would like to thank the authors for their appreciation of my review and for incorporating the suggested changes. I think the authors did a great job in revising their paper. Most of my comments have been addressed, but I still have some minor comments below. I especially appreciate the new information on the BAPPS dataset, as well as the new experiments on the skip connection and the analysis on the relationship to the entropy as a possible source of decreasing PS score. I also think the results section looks much better and more comprehensive. I appreciate the authors fixing the y-ranges in all plots.
> >
> > - Figure 3. Thanks for adding it! It is not really clear to me what "Scale" means as the x-axis here. Could you please explain the word "scale" in the description next to the Figure or rephrase it somehow?
> >
> > - Figure 5: Figs. 5a and 5c show PS vs accuracy as a function of train epochs in ResNets and ViTs. Figs. 5b
> > and 5d depict PS as a function of train epochs. -> I believe the "as a function of train epochs" in a+c is a typo. I have not checked all the Figure captions for typos; please make sure that there are none, i.e. check all Figures+Tables for typos.
> >
> > - Formatting around Fig. 17 is off, please fix.
> >
> > - Results on the activation entropy (section 8.4).
> >   - Figure 14b suggests that there is a very clear relationship between activation entropy and PS score if one excludes the checkpoint at initialization (I believe this is the outlier?) which sounds sensible to me. Is 14b plotted for the 5 different seeds or are these the averaged results? I find this plot surprising since 14a clearly shows that entropy goes down quite monotonically during training while this does not happen for the PS score, so how come there is such a clear relationship between the PS score and entropy in Fig. 14b?
> >   - The variance between the different seeds in Figure 14a is really big, and suggests that one probably cannot relate entropy to anything else if the variance is this big in all entropy-related experiments. Given this large variance, I am actually surprised by the clear relationships shown in 14b and 14c. Checking Figure 4, the error bars on both the PS score and accuracy are really small, so the results on these two metrics do not depend on the seed in contrast to the activation entropy.
> >   - The observation that there is no relationship between different models can be explained by the big within-model variance. I think given the large differences between different seeds (discussed above), one can only look at the PS score and entropy within one model across training, but one cannot compare the entropy between two different models, even from the same architecture. (I believe [Wang et al 2021] have actually not reported results across different seeds and do not show any error bars in their paper, so seeing Fig.14a, I actually no longer trust the findings from [Wang et al 2021]. This goes beyond the discussion on this paper though. Although the authors may want to comment on [Wang et al 2021] and the possible issues related to the strong variance in activation entropy).
> >   - Given these arguments, I think it would make more sense to color the points in Fig 4b using the colors from Fig 14a (and thus identify the different models) since plotting entropy across different models does not make sense due to the large variance.

---

> > > ### Author Response · Authors · 2022-07-29
> > > **Revision**
> > >
> > > Thanks for the quick response and helpful feedback. We appreciate it! We uploaded a revision draft responding to this feedback.
> > >
> > > * Figure 3: We renamed the x-label to $\sigma$. We added "$\sigma$ models the noise in predicting the simulated distances from the real labels" to the caption.
> > >
> > > *  We changed "PS vs accuracy as a function of xyz" to "PS vs accuracy on varying xyz" in all captions.
> > >
> > > * Fig. 17 is unfortunately an artifact of using wrapfig at the end of the page. We will move this subsection around toward the end of the discussion phase.
> > >
> > > * **Entropy Experiments** - Each colored line in Fig, 14 a) is not a random seed but the output of a different reduction stage in ResNet, that is one of the 4 2-D representations use to compute the PS. We made the following changes to clarify this in the draft.
> > >   * Legend now says "Reduction Stage: 1" instead of 1
> > >    * Caption now says: "Fig. 14a depicts the entropy of each of the four ResNet reduction stages as a function of train epochs. The black line is the mean activation entropy across the 4 reduction stages"
> > >    * Text now reads: Fig. 14a plots the normalized entropy of each of the four ResNet-200 reduction stages (marked 1 - 4) across
> > > training. As seen in Wang et al. (2021a), representations closer to the output at later reduction stages have
> > > a much lower entropy than representations closer to the input at earlier reduction stages.
> > >
> > > Given that Fig 14 a) does not show the variance in seeds but rather in variance across the ResNet stages, hopefully the other comments are also resolved.
> > >
> > > * **Entropy Experiments** - For ResNet-200, the PS peaks very early training after 2-3 epochs (Fig. 5b) and then monotonically decreases. One more point is not visible in 14b) due to the y-axis limit at 67.0
> > >
> > > Let us know if you have other questions.

---

> > > > ### Comment · Reviewer_7meX · 2022-07-29
> > > > **Question on the entropy experiments**
> > > >
> > > > Ah I see thank you. Yes, the issues are resolved then. In 14b, the correlation is still quite significant. You write "While there is an overall correlation (both entropy and PS decrease throughout training), the maximum entropy is at initialization and not after a few training epochs where ResNet-200 obtains its highest PS. Therefore, entropy does not explain fully the observed effect." I get your objection towards using entropy to explain this effect, but it still seems to be quite important, right? Maybe at initialization, the representations are too "random and chaotic" and therefore not useful. Then, they become more class-specific, which makes them more useful, but they also lose entropy, and thus lose their representational capacity. So there may be a sweet-spot in terms of class-specific representations versus entropy. There is a paper on contrastive learning which tries to understand which augmentations to use and in what strength (https://proceedings.neurips.cc/paper/2020/hash/4c2e5eaae9152079b9e95845750bb9ab-Abstract.html). They also observe an inverse V-shape when plotting augmentation strength against accuracy and use information-theoretic arguments to explain their findings. I wonder whether some similar arguments could be applied here. If their framework is not applicable here, let me know, that's completely fine of course.

---

> > > > > ### Author Response · Authors · 2022-08-01
> > > > > **Entropy Experiments: Discussion**
> > > > >
> > > > > Thanks for engaging in the discussion! We uploaded a new revision with some changes.
> > > > >
> > > > > * The positive correlation in Fig. 14 b) is also an artifact of logging every 1000 training steps. In the new Fig 14 b), for the first few training epochs, we plot every 50 training steps. Here we see, that the negative correlation between PS and entropy is more pronounced in the high entropy regime, i.e between an entropy of 0.9 to 1.0, there is a range of PS from 67.0 to 70.0
> > > > >
> > > > > * As rightly observed, it does seem there is a "entropy sweet spot" as a function of train steps, where the PS peaks. However we are hesistant in making general claims about the effect of entropy beacuse of the following counter-examples:
> > > > >
> > > > >   * Fig 14 d) shows no sweet spots or correlation in entropy vs PS when depth is varied.
> > > > >   * When skip connections are removed, entropy increases across all depths, which seems to help in style transfer. However, PS consistently decreases across depth across a wide range of entropies.
> > > > >
> > > > > We reworded some parts of Section 8.4 slightly, highlighting the "optimal entropy" as a function of training in response to this discussion:
> > > > >
> > > > > -----
> > > > > For the first few training epochs, where the entropy is between 0.9 and 1.0, there is a negative correlation between PS and mean activation entropy. Note that the maximum entropy is at initialization and not after a few training epochs where ResNet-200 obtains its highest PS. Afterthe first few epochs, there is a positive correlation where entropy and PS both decrease during training. Fig. 14a suggests that there is a optimal entropy as a function of train steps, where the PS peaks. However, Fig. 14d shows that there is almost no correlation between entropy and PS across ResNets with various depths. ResNet-6 achieves the highest PS at a mean entropy of≈0.5 while ResNet-50 features have the highest entropy around 0.7 and have a much lower PS of 68.0. Therefore, entropy does not fully explain the observed effect
> > > > >
> > > > > ----
> > > > >
> > > > > The paper that you linked, unlike the style transfer paper, falls outside our framework, since it is not obvious to tractably define a notion of MI between the PS and classification task.
> > > > >
> > > > > Please let us know if the current draft sufficiently addresses all your concerns or if you have any further questions. Thanks!

---

> > > > > > ### Comment · Reviewer_7meX · 2022-08-02
> > > > > > **Response to authors about the activation entropy**
> > > > > >
> > > > > > Thank you for plotting the 14b graph in better scale, now it is definitely more visible that entropy does not explain the studied effect.
> > > > > >
> > > > > > One more note: The authors write " For example, ResNets trained with large amounts of weight decay or early stopped within a few epochs of training can match or outperform the PS of AlexNet and VGG." But I could not find the PS score number for VGG in the paper, please add it to Figure 3 and to the text next to Figure 3.
> > > > > >
> > > > > > I do have one more suggestion: Since the authors have studied how various architecture choices and number of training steps affect the perceptual score in great detail, would it be possible or advisable to formulate a set of rules or suggestions which researchers should follow when evaluating perceptual similarity? This paper could shift the way how researchers evaluate perceptual distance in the future. For this, there needs to a set of well-motivated rules which steps need to be followed. Is it advisable to use an early stopped ResNet-6 for example, rather than AlexNet or VGG? I think with such a set of rules, this paper could reach higher visibility and traction within the community. This is a minor suggestion though which may or may not be a good idea. I would definitely be interested to hear the authors' thoughts on it, even if it is not advisable to formulate such a set of rules in the paper.
> > > > > >
> > > > > > Otherwise, the paper looks solid to me :).

---

> > > > > > > ### Author Response · Authors · 2022-08-04
> > > > > > > **Checkpoints Release**
> > > > > > >
> > > > > > > Thanks for the positive feedback and the suggestion!
> > > > > > >
> > > > > > > For the final version, we will release the checkpoints for the models that achieve the best Perceptual Scores. Our recommendation would not  be to replace AlexNet/VGG completely with the checkpoints from our paper but moreso to offer an alternative, high-performing metric for perceptual evaluation.
> > > > > > > This, for instance would help the community to not overfit to a single perceptual metric based on AlexNet/VGG.

---

### Review · Reviewer_9e1v · 2022-07-24

**Summary Of Contributions:**

The paper explores the relationship between ImageNet accuracy and Perceptual Scores of modern deep models such as ResNets, EfficentNets, and Vision Transformers, and found they are inferior in Perceptual Scores. Through a series of ablation studies on hyperparameters, such as depth, width, number of training steps, weight decay, and label smoothing, the paper further discover some interesting observation such as shallow, early-stopped ResNets has a surprisingly good Perceptual Score of 70.2.

**Requested Changes:**

See weakness above

**Strengths And Weaknesses:**

Strengths:
(a) The topic of the relationship between ImageNet accuracy and Perceptual Scores is worth exploring and could be helpful to the community.
(b) The paper is overall well-organized and the writing quality is pretty good.
(c) The paper is packed with experiments, the author provides a lot of ablation studies on hyperparameters (e.g. depth, width, number of training steps, weight decay, label smoothing), w.r.t perception scores.
(d) The paper further explores the more advanced setting, such as different Global Similarity Functions, Distortion Sensitivity, Layer-wise Perceptual Scores, Spatial Frequency, Sensitivity, and ImageNet Class Granularity, that also brings value to the community.

Weaknesses:
(a) The paper provides a lot of ablation studies In Sec. 4/5/6, however, lacks some insight on WHY those observations should happen in a neural architecture perspective. for example, (i) In Figure 2, ResNet, EfficientNet, ViT all have "skip" passes, Could that be the reason why they turned out better in classification and less effective in perception scores? (ii) Shallow, Early-stopped ResNets has a good Perceptual Score compared to other settings, is it because Shallow ResNets are easier to train?.
(b) Why there is no similar ablation on EfficientNets in Figure 3?
(c) minor: legend in Figure 2, Efficient -> EfficientNet, Figure 6: Number of ImageNet classes trained on XXX?

---

> ### Author Response · Authors · 2022-07-28
> **Response**
>
> Thanks for the thoughtful review. See below for our response. All changes are in the manuscript in colour **NavyBlue**.
>
> ---
> ### Role of skip connections
> R-7mex has the same suggestion. We added Section 8.4 exploring the relationship between the entropy of activations and PS. We additionally removed skip connections from ResNets and show that the inverse-U persists. Here we describe the Section.
>
> [Wang et al 2021] show that ResNets are not suitable for style transfer because of skip connections. Skip connections result in features with low entropy, which in turn prevent capturing all style modes from the ground-truth style image. We now explore if features with lower entropy can explain the inverse-U phenomenon. As done in [Wang et al 2021], we convert  into a probability distribution across  values by applying a softmax transformation. We then report the average normalized entropy of this distribution across the 4 representations.
>
> Fig. 14 a plots the normalized entropy of each ResNet-200 representation across training. As seen in [Wang et al 2021] representations at later stages have a much lower entropy than earlier stages. The entropy also decreases as a function of train steps. However, the maximum entropy is at initialization and not after a few training epochs where ResNet-200 obtains its highest PS. Further, Fig. 14 d shows that there is almost no correlation between entropy and PS across ResNets with various depths. ResNet-6 achieves the highest PS at a mean entropy of  while ResNet-50 features have the highest entropy around 0.7 and have a much lower PS of 68.0.
>
> We then remove all skip connections in the ResNet models to increase the entropy of the intermediate features as done in [Wang et al 2021]. Note that the maximum depth that we are able to successfully train is 50. In Fig. 15 a, removing the skip connections, increases the entropy across all depths and especially ResNet-50, which has an entropy close to 1.0. Even after removing the skip connections, we see in Fig. 15 b and Fig. 15c that ResNet-6 and early-stopped ResNets attain the highest PS similar to the baselines
>
> Note that the ResNets without skip connections are quite similar to VGG networks module the ordering of convolutions, max pooling layers and batch norm layers, which are shown not to have a significant effect on style transfer in [Wang et al 2021]. We have provided sufficient evidence that skip connections do not explain the inverse-U phenomenon.
>
> -----
>
> ### EfficientNet ablations
>
> We added some ablations in Appendix I for EfficientNet-B0 and EfficientNet-B5. We observe the inverse-U tradeoff in EfficientNet-B0 and EfficientNet-B5 as a function of depth,width and number of epochs respectively. EfficientNets attain their peak PS at extremely small width, depth and number of train steps. We omit weight decays (similar to ViT-L/4) since EfficientNets are fairly robust to the choice of weight decay parameters.
>
> ----
>
> ### Stability in Training
>
> * For ResNets, we rely on standard and established training recipes. We do not observe a lot of fluctation in the training losses across time or final accuracies for both shallow and deeper ResNets.
> *  PS can also be increased via hyperparameters other than depth. For example, central cropping which arguably should not have a huge influence on training stability.
>
> ----
>
> ### Other
> * minor: legend in Figure 2, Efficient -> EfficientNet (fixed)
> * Figure 6: Number of ImageNet classes trained on XXX? -> We renamed this to "Number of classes" and expanded the caption instead. The caption now states: "We create ImageNet subsets by randomly sampling x number of classes and then train models on
> each of these subsets. The plot shows PS of ResNet-6 and ResNet-200 models vs number of ImageNet classes
> (x) in these subsets."
>
> ----
>
> We believe we have addressed all your comments and concerns and look forward to your updated impression of our paper.

---

### Review · Reviewer_KGpd · 2022-07-25

**Summary Of Contributions:**

This paper investigates the relationship between the accuracy of an image classification network and its ability to predict perceptual similarity as judged by humans. Previous work had identified a positive correlation between the accuracy of a network and its ability to predict perceptual similarity, but this work finds that the opposite is true for more modern architectures. The relationship is explored over a variety of architectures and hyperparameters and the results indicate that there is an inverse-U relationship between accuracy and perceptual score (PS): as the accuracy improves the PS increases up until a point and then decreases again.

Contributions include:
- Experiments that characterize the relationship between accuracy and perceptual score for modern architectures.
- Hyperparameters sweeps that demonstrate the effect thtat changes to the architecture and training procedure have on the PS.
- Proposal of an approach to improve PS, namely to reduce the scale of the model until the optimal (lower) accuracy is achieved.

**Broader Impact Concerns:**

Discussion could be added on how the results of this paper depend on the underlying perceptual similarity dataset and the suitability of a global perceptual similarity score. In particular, might it be the case that the conclusions of this paper depend on internal biases of the human subjects whose similarity judgments were collected to form the dataset? Is it appropriate to target a single global perceptual similarity score, or might it be the case that different individuals perceive images differently? If so, what might the implications be of using a neural network to model perceptual similarity in real-world applications?

**Requested Changes:**

- Improve the motivation for perceptual similarity and how better perceptual similarity could have a positive impact.
- Include more discussion about the reasons for the inverse-U relationship, ideally supported by theory and experiments.
- Improve presentation of the figures and include missing labels in Figures 3 and 4.
- Limitations should be discussed in greater detail. In particular, two points come to mind:
  1. To what extent are these results predicated on the nature of the BAPPS dataset? If there were a hypothetical dataset that had collected perceptual similarity judgments on higher resolution images, e.g. 224 x 224, how would the results and conclusions of this paper change?
  2. What are the limitations of the distance function (eq 1) and scoring function (eq 2)? The scoring function only cares if $d_0 > d_1$ or vice versa, but is there any merit to considering a scoring function where the network should predict $p$ directly, rather than simply which distance is larger?

**Strengths And Weaknesses:**

Strengths
- The experiments are systematic, including modern architectures like ViT and sweeps over a large number of hyperparameters.
- With the aim of improving the PS, a concrete suggestion is proposed: reduce the scale of the model.
- Discussion of the hyperparameter sweeps helps to shed insight into the relationship between accuracy and PS.

Weaknesses
- The takeaway message is somewhat unclear. The experiments do a good job of establishing the relationship between accuracy and PS, but more could be done to motivate the problem of trying to improve PS and whether a higher PS is intrinsically useful.
- The discussion and analysis of the results are relatively light. The findings seem to indicate that weaker models are better for PS, but there is little discussion of the deeper reasons for this. For example is the inverse-U phenomenon related to overfitting, lack of calibration, lack of biological plausibility, etc.?
- Some of the figures are difficult to parse, especially Figure 6.

---

> ### Author Response · Authors · 2022-07-29
> **Response (1 of N)**
>
> Thanks for the thoughtful review! See below for our response. All our changes are incorporated in the manuscript with color **ForestGreen**
>
> -----
> ### Scoring Function: Limitation
>
> We believe our results should be agnostic to the scoring function: i.e
>
> a) Which distance is larger? $1[d_1 > d_0]$ or
>
> b) The absolute distance margin itself. $(d_1 - d_0)$
>
> We added the following experiments to **Appendix J** and **Fig. 30** to empirically verify this
>
> The scoring function (Eq. 2) takes into account only if $d_1 > d_0$, and not the actual margins ($d_1 - d_0$) themselves. To assess if the inverse-U relationship is agnostic to this choice, we use a scoring function that takes the distance margins into account. Namely, we measure the rank correlation between the distance margins ($d_1 - d_0$) and the ground truth probabilities $p$ in BAPPS. A model having a high rank correlation should assign large positive distance margins at high probabilities and large negative distance margins at low probabilities. In Fig. 30, we show that similar to PS, rank correlation exhibits an inverse-U relationship for a) ResNets at different depths b) ResNet-200 as a function of train epochs. Therefore the conclusion remains unchanged.
>
> -----
>
> ### BAPPS Resolution: Limitations
>
> We also believe that our observations should be agnostic to the resolution of images tested.  We add the following piece of evidence in **Appendix K** and **Fig. 31**
>
> BAPPS was explicitly constructed using $64×64$ images so that human raters can focus on low-level local changes as opposed to high-level semantic differences. We show results on the TID2013 dataset (Ponomarenko et al., 2015), which consists of higher resolution images ($384 \times 512$). Each example in TID2013 consists of a reference image $x$, a distorted image $y$ and a mean opinion score rating $s$. For every model, we evaluate the rank correlation between $s$ and $d(x,y)$ (as given in Eq. 1). Firstly, we assess ResNets of various depths trained on ImageNet $224×224$. Secondly, we estimate the rank correlation of a ResNet-50 model trained on ImageNet $384 \times 384$ during the course of training. Fig. 31b and 31d show that shallow ResNets and early-stopped ResNets achieve the highest rank correlation. In Figs. 31a and 31c, we see that the inverse-U relationship exists on TID2013.
>
> Note that TID2013 consists of only traditional distortions as compared with BAPPS which has a richer set of both traditional and neural distotions. So, our analysis in the main section is exclusively focused on BAPPS.
>
> -----
>
> ### Improve presentation of the figures .
>
> We have made several changes to improve the readability of plots:
>
> * We moved each plot in (previous) Figure 3 and Figure 4 to under the corresponding paragraphs. For example: Figure 5 is placed under “Number of Train Epochs” and contains 4 plots with the caption “Figs. 5a and 5c show PS vs accuracy as a function of train epochs in ResNets and ViTs. Figs. 5b and 5d depict PS as a function of train epochs.”
> * Since ViTs and ResNets have different dynamic ranges of PS, we decoupled these plots and made the y-axis consistent for each architecture across all subplots. For ResNets the y-axis is between 67.0 and 70.0. For ViTs, the y-axis is between 66.0 and 69.0
> * We moved each plot in (previous) Figure 6 to the corresponding sub-sections. For example: Figure 13 is placed under Section 8.3 and the caption of Figure 13 reads "Per-layer PS" for ResNets with different depths in Fig. 13a) and for an early-stopped ResNet-200.
>
> ----
>
> ###  Perceptual Similarity Motivation
> Designing distance metrics that correspond better to human judgements is a long standing problem in computer vision, even before the advent of deep learning (see: SSIM and FSIMc). Analyzing representations of ImageNet classifiers is also receiving increasing attention over the past few years. Our paper bridges these two fields. Our paper is not strictly only about improving PS, but also to provide a comprehensive analysis between the accuracy of ImageNet classifiers and perceptual similarity. Our paper is also of interest to the representation learning community, given that we study the transfer of ImageNet representations to a different domain.
>
> We added the following in the **introduction** to improve the high-level motivation.
> * In this paper, we are motivated by the following questions: Considering ImageNet classification has progressed significantly since then, can we obtain a better perceptual similarity metric by using a better classifier directly? Since modern neural network training involves a large number of hyperparameters, are there design choices that can improve a classifier’s perceptual similarity? Are there latent factors that govern the relationship between ImageNet accuracy and perceptual similarity?
> * Given the increasing interest in analyzing how representations of ImageNet classifiers transfer to other domains, our work adds another direction to this literature.

---

> ### Author Response · Authors · 2022-07-29
> **Response 2 of N**
>
> ### Reasons for the inverse-U relationship
> In **Section 6**, we explore plausible reasons for this relationship. We renamed this section from **Further Exploration** to **Further Exploration: Reasons behind the inverse-U relationship** to be more explicit. Following comments from 7Mex and 9e1v, we added a  subsection that investigates the impact of skip connections. We renamed each subsection directly to reflect the conclusion of the hypotheses. We restate each hypothesis that we tested here and the conclusions:
>
> * **Section 8.1**: Does the inverse-U persist with global perceptual functions?
>   **Experiments**: We investigate 2 “global perceptual functions” (mean-pool and style)
>   **Results**: Inverse-U exists in 1) ResNet-200 as a function of training epochs. Fig. 11 a) 2) ResNets, EfficientNets and ViTs trained with their default hyperparameters. Fig. 11 b)
>
> * **Section 8.2**: Are low PS models less sensitive to distortions?
>   **Experiments**: We examine the relationship between the average distance margin on examples with unanimous ratings ($p=0.0$ and $p=1.0$) and PS.
>   **Results**: We show that there is no correlation between the sensitivity of models to distortions and PS in both 1) ResNet-200 as a function of training epochs Fig. 12 a) 2) ResNets, EfficientNets and ViTs trained with their default hyperparameters. Fig. 12 b)
>
> * **Section 8.3**: Do lower layers of better classifiers have a higher PS than higher layers?
>   **Experiments**: We examine the layer-wise PS to see if the “optimal” representations for classifiers with different PS are in different layers.
>   **Results**: We show that the “optimal layer-wise PS” for a) ResNet-6 is higher than ResNet-200 b) Early-stopped ResNet is higher than ResNet-200
>
> * **Section 8.4**: What are the contributions of skip connections in modern architectures to decreased PS, if any?
>   **Experiments**: Previous work on style transfer, showed that skip connections can cause low entropy features, which make ResNets unsuitable for style transfer. We study the relationship between entropy and PS.
>   **Results**: We show that a) Entropy has no correlation with PS across depths Fig. 14 c) b) Inverse-U exists even after removal of skip connections. Fig.15 b and 15 c
>
> * **Section 8.6**: What is the impact of ImageNet class granularity on PS?
>   **Experiments**: Since Imagenet is a 1000 class category, focussing on fine-grained classes, could compromise on learning general features, which could lead to a decreased PS.
>   **Results**: We create ImageNet subsets with reduced classes and train ResNet-6 and ResNet-200 models on each subset. In Fig. 17, the gap in PS between ResNet-6 and ResNet-200 still exists across each subset. In Appendix E, we show similar results with subset selection strategies guided by a pretrained ResNet-6.
>
> **Overfitting**: In Appendix H, we showed the validation accuracy against hyperparameters, to demonstrate that the networks do not overfit on ImageNet. Note that the peak PS is far away from where the validation accuracy starts to plateau: (extremely small widths, depths, early stopped after the first few epochs, large amounts of weight decay).
>
> **Calibration**: Though we did not consider calibration explicitly in this paper, prior work indicates that with modern ImageNet networks, better classifiers are tend to be better calibrated also [1]. In the contrary, our work demonstrates that moderate/weak classifiers achieve the best PS.
>
> We acknowledge that these results do not explain the cause of the inverse-U relationship and are transparent about this claim (Page 2, paragraph 3); however, these are very useful to the community to rule out these hypotheses/directions. Note that the reason why ImageNet classifiers are able to exhibit perceptual similarity as an emergent property as observed in Zhang et al. [2018] is unclear. As with most new empirical findings in deep learning, a theoretical explanation is far beyond the scope of the paper, but we hope that the theory community will follow up with further analysis.
>
> ----
>
> ### Broader Impact Statement
>
> We framed this broader impact statement.
>
> Our results are based on BAPPS, which consists of exclusively low-level distortions as opposed to high-level semantic differences. We believe low-level distortions such as gaussian blur and color distortions are less likely to be susceptible to bias across different human categories as compared to high-level semantic features such as facial features. It is an open and interesting question whether different categories of humans like race and gender perceive low-level distortions differently. Increasing the diversity of both distortions and human labels in future perceptual similarity datasets is another interesting direction that might help to mitigate human biases.
>
> ---
> We believe we have addressed the concerns and comments. We look forward to reading your updated opinion of the paper.
>
> [1] Revisiting the Calibration of Modern Neural Networks [Minderer et al. 2021]

---

### Author Response · Authors · 2022-10-29
**Checkpoints release**

Hello all,

We released the checkpoints of the best performing models in the paper for ResNet-6, ResNet-50 and ResNet-200 over here (https://console.cloud.google.com/storage/browser/gresearch/perceptual_similarity). See here also for a write-up (https://ai.googleblog.com/2022/10/do-modern-imagenet-classifiers.html)

Thanks again!

---

### Decision · Action_Editors · 2022-09-07

**Recommendation:** Accept as is

**Comment:**

The paper is reviewed by three expert reviewers in the field. Initially, the reviewers have concerns with the clarity of the experimental sections and unclear motivations. Reviewers KGpd and 7meX appreciate the detailed responses by authors and their efforts on revising the manuscript. The quality and clarity of the revised paper has been significantly improved with strong empirical results on the studied phenomenon.

Both Reviewers KGpd and 7meX expressed that the authors have done a great job addressing all their major concerns and recommend to accept as is. While Reviewer 9e1v did not update the final recommendation, but after reading the authors' responses, the AE agrees that the authors revision sufficiently addressed the concerns. In particular, the added portion in Section 8.4 that explore the relationship between the entropy of activations and PS and the newly included ablation for EfficientNet-B0 and EfficientNet-B5 in Appendix I.

The AE agrees with the reviewers and recommends to accept as is. Congratulations!

---

> ### Author Response · Authors · 2022-09-08
> **Camera-Ready version**
>
> Dear reviewers and AE
>
> Thanks for the time and energy invested in the review process, that improved the quality of our paper. We uploaded the deanonymized camera-ready version.